# The genomic landscape of transposable elements in yeast hybrids is shaped by structural variation and genotype-specific modulation of transposition rate

**Mathieu Hénault[1,2,3,4]\*, Souhir Marsit[1,2,3,4,5]†, Guillaume Charron[1,3,4,5]‡, Christian R Landry[1,2,3,4,5]**

[1]Institut de Biologie Intégrative et des Systèmes (IBIS), Université Laval, Québec, Canada; [2]Département de biochimie, microbiologie et bioinformatique, Université Laval, Québec, Canada; [3]Quebec Network for Research on Protein Function, Engineering, and Applications (PROTEO), Université Laval, Québec, Canada; [4]Université Laval Big Data Research Center (BDRC_UL), Québec, Canada; [5]Département de biologie, Université Laval, Québec, Canada

**\*For correspondence:**
mathieu.henault.1@ulaval.ca

**Present address:** †Département de biologie, chimie et géographie, Université du Québec à, Rimouski, Canada; ‡Centre de foresterie des Laurentides, Ressources naturelles Canada, Québec, Canada

**Abstract** Transposable elements (TEs) are major contributors to structural genomic variation by creating interspersed duplications of themselves. In return, structural variants (SVs) can affect the genomic distribution of TE copies and shape their load. One long-standing hypothesis states that hybridization could trigger TE mobilization and thus increase TE load in hybrids. We previously tested this hypothesis (Hénault et al., 2020) by performing a large-scale evolution experiment by mutation accumulation (MA) on multiple hybrid genotypes within and between wild populations of the yeasts *Saccharomyces paradoxus* and *Saccharomyces cerevisiae*. Using aggregate measures of TE load with short-read sequencing, we found no evidence for TE load increase in hybrid MA lines. Here, we resolve the genomes of the hybrid MA lines with long-read phasing and assembly to precisely characterize the role of SVs in shaping the TE landscape. Highly contiguous phased assemblies of 127 MA lines revealed that SV types like polyploidy, aneuploidy, and loss of heterozygosity have large impacts on the TE load. We characterized 18 de novo TE insertions, indicating that transposition only has a minor role in shaping the TE landscape in MA lines. Because the scarcity of TE mobilization in MA lines provided insufficient resolution to confidently dissect transposition rate variation in hybrids, we adapted an in vivo assay to measure transposition rates in various *S. paradoxus* hybrid backgrounds. We found that transposition rates are not increased by hybridization, but are modulated by many genotype-specific factors including initial TE load, TE sequence variants, and mitochondrial DNA inheritance. Our results show the multiple scales at which TE load is shaped in hybrid genomes, being highly impacted by SV dynamics and finely modulated by genotype-specific variation in transposition rates.

## eLife assessment

This **valuable** study advances our understanding of the forces that shape the genomic landscape of transposable elements. By exploiting both long-read sequencing of mutation accumulation lines and in vivo transposition assays, the authors offer **compelling** evidence that structural variation rather than transposition largely shapes transposable element copy number evolution in budding yeast. The work will be of interest to the transposable element and genome evolution communities.

DOI: https://doi.org/10.7554/eLife.89277

**Figure 1.** Examples of how structural variants (SVs) can impact the genomic landscapes of transposable elements (TEs) in hybrids. An ancestral heterozygous genome is shown as an example, with two subgenomes (black and gray) harboring TEs (green boxes) in distinct copy numbers and at different insertion loci. Examples of various types of SVs are shown in derived genomes: polyploidy (gain of haploid sets of chromosomes), aneuploidy (gain or loss of whole chromosomes), loss of heterozygosity (LOH, inter-homolog conversion of chromosome segments), and de novo transposition. The corresponding changes in TE copy numbers are shown on the right.

## Introduction

Structural variants (SVs) are a powerful source of genetic variation that fuel the evolution of genomes and species (*Gorkovskiy and Verstrepen, 2021*). Structural variation is a term that encompasses a broad variety of large-scale sequence alterations that can have profound consequences for organismal phenotypes and genome evolution. For example, aneuploidies can underlie the emergence of anti-fungal resistance in fungal pathogens (*Selmecki et al., 2006*) and drive tumorigenesis in mammalian cells (*Shoshani et al., 2021*; *Trakala et al., 2021*). Polyploidization is frequently linked to speciation and adaptation, notably in plants (*Otto and Whitton, 2000*). Mitotic recombination can yield loss of heterozygosity (LOH) events that drive adaptation in heterozygous hybrid genomes (*Smukowski Heil et al., 2017*), but also the somatic fixation of oncogene mutations (*Cavenee et al., 1983*). Repeated sequences are known to play a leading role in the generation of SVs (*Lower et al., 2019*; *Todd et al., 2019*). Notably, transposable elements (TEs) are sequences capable of semi-autonomous replication, producing families of repeats spread within genomes (*Bourque et al., 2018*). Because of their replicative nature, TEs are major generators of SVs: not only do they create interspersed duplications of themselves, but they can also trigger genome rearrangements through ectopic recombination (*Dunham et al., 2002*; *Mieczkowski et al., 2006*; *Startek et al., 2015*). While TEs can cause SVs, various types of SVs can also affect the genomic landscape of TEs by producing gains and losses of individual copies (*Figure 1*).

TEs are inherent components of virtually all eukaryotic genomes and owe their evolutionary success to their own replicative ability. However, this selfish propagation conflicts with the fitness of the host and can have profound deleterious impacts (*Bucheton et al., 1984*; *Pasyukova et al., 2004*; *Wilke and Adams, 1992*). The abundance of TEs per genome (or TE load) is an important population genetic parameter as it is expected to correlate with the fitness cost for the host. TE load in a population is most often modeled as a quantity reflecting the balance between the gain of novel copies by transposition and the decrease in copies by loss (e.g., by excision or loss of function through

mutation) and purifying selection against insertion alleles (*Charlesworth et al., 1994*; *Charlesworth and Langley, 1989*). However, TE load is not only defined by the copy number (CN) of TEs in the genome, but also where TE insertions fall and, consequently, how they interact with neighboring host sequences. The multidimensional nature of TE load makes its full characterization a complex task. Recent advances in long-read sequencing make it increasingly possible to decompose TE load by resolving the genomic context specific to individual insertion loci (*Oggenfuss and Croll, 2023*; *Rech et al., 2022*). Additionally, accurate genome resolution opens unprecedented opportunities for a comprehensive assessment of the role of SVs in shaping TE load. As such, ongoing technological and computational advances in genome inference, including long-read sequencing, will certainly be key to getting a detailed understanding of the dynamics of TEs and the underpinning evolutionary forces.

One long-standing hypothesis postulates that hybridization can cause a genomic shock that triggers TE reactivation through increased mobilization (*McClintock, 1984*), with the resulting elevated load having major potential implications for speciation and hybrid evolution. This hypothesis was tested in many systems, some of which provided evidence for reactivation (*Dion-Côté et al., 2014*; *O'Neill et al., 1998*; *Ungerer et al., 2006*). In some cases, the prevailing mechanism of TE repression by the host explained the molecular basis of reactivation. For example, in *Drosophila melanogaster*, the absence of maternally inherited piRNAs is responsible for the failed epigenetic repression of the *I* and *P* elements, leading to the expression of a hybrid dysgenesis syndrome (*Brennecke et al., 2008*). However, this level of mechanistic understanding is not often matched in other systems. The reactivation hypothesis remained untested in fungi until recent works in budding yeasts of the genus *Saccharomyces* and in the fission yeast *Schizosaccharomyces pombe* (*Drouin et al., 2021*; *Hénault et al., 2020*; *Smukowski Heil et al., 2021*; *Tusso et al., 2022*).

In the model species *Saccharomyces cerevisiae*, the only TEs found are five families of long terminal repeat (LTR) retrotransposons named Ty1-Ty5 (*Kim et al., 1998*). Its undomesticated sister species *Saccharomyces paradoxus* comprises the related families Ty1, Ty3, and Ty5 (*Yue et al., 2017*), in addition to Tsu4, which was horizontally transferred from *Saccharomyces uvarum* (*Bergman, 2018*). *Saccharomyces* species lack most of the defense lines typical of other eukaryotes, like DNA methylation and RNA interference (*Bewick et al., 2019*; *Drinnenberg et al., 2009*). The best-known regulation mechanism in yeast is termed copy number control (CNC) and was characterized in the Ty1 family of *S. cerevisiae*. This mechanism is a potent copy-number-dependent negative feedback loop by which increasing the CN of Ty1 elements strengthens their repression (*Czaja et al., 2020*; *Garfinkel et al., 2003*; *Saha et al., 2015*). CNC was shown to regulate the mobilization of one subfamily of Ty1 elements in *S. cerevisiae*, but whether CNC is active in other species remains unknown. Additionally, systematic gene deletion studies revealed that multiple cellular functions can restrict or promote Ty1 transposition (reviewed in *Curcio et al., 2015*), setting the stage for potential genetic incompatibilities to disrupt the regulation of retrotransposition.

We previously tested the reactivation hypothesis in a wide diversity of *Saccharomyces* hybrids by performing a large-scale evolution experiment by mutation accumulation (MA) to investigate the near-neutral evolution of TEs in hybrid genomes (*Hénault et al., 2020*). We generated a collection of MA lines comprising a set of 11 hybrid genotypes created by crossing wild strains from North American populations of *S. paradoxus* and *S. cerevisiae* (*Charron et al., 2014a*; *Kuehne et al., 2007*; *Leducq et al., 2014*; *Leducq et al., 2016*) that are separated by a range of evolutionary divergence and by replicating each cross many dozen times independently (*Charron et al., 2019*; *Hénault et al., 2020*). The resulting collection was evolved by subjecting each line to periodical extreme bottlenecks by streaking for single colonies on rich medium for ~770 mitotic generations, thus amplifying the power of random genetic drift. We previously sequenced a subset of these MA lines with short reads and measured the change in Ty load after evolution by producing aggregate estimates of relative abundance for each family and found no support for the TE reactivation hypothesis (*Hénault et al., 2020*). However, the use of short reads alone precluded the detailed resolution of hybrid genomes and the decomposition of their Ty load. For instance, different SVs (including de novo transposition) may have affected Ty CN in opposite directions. Under this scenario, measuring Ty family abundance would yield no significant net change, and the dissection of the underlying SVs using short reads could often be challenging.

While producing collapsed assemblies for haploid or predominantly homozygous genomes is relatively straightforward, complex heterozygous genomes are much more difficult to characterize at the

haplotype level. Recent studies demonstrated the feasibility of the approach (*Heasley and Argueso, 2022*; *O'Donnell et al., 2023*), but the task of resolving hybrid genomes across a wide range of parental divergence under a unified framework remains challenging. Here, we employ a phasing and assembly approach using long reads to resolve hybrid MA lines genomes. We find that de novo transposition events play a minor role in shaping the Ty load in comparison to other SV types. Since the number of de novo insertions gained during MA is low, we further dissect transposition rate variation in *S. paradoxus* hybrids at a higher resolution with in vivo transposition rate measurements. We identify many aspects of hybrid genotype specificity that shape transposition rates, including initial Ty load, Ty sequence variation and mitochondrial DNA (mtDNA) inheritance.

## Results

The 11 MA crosses (*Charron et al., 2019*; *Hénault et al., 2020*) comprise five crosses within *S. paradoxus* populations (CC, within *SpC*; BB, within *SpB*), four crosses between *S. paradoxus* populations (BC, *SpB×SpC*; BA, *SpB×SpA*), and two interspecific crosses (BSc, *S. paradoxus SpB×S. cerevisiae*) (*Figure 2A*). We previously selected 127 independently evolved MA lines (10–12 per cross) and sequenced their genome with Oxford Nanopore long reads (*Hénault et al., 2022*). To produce an accurate representation of the two distinct subgenomes of each evolved MA line, we employed a strategy consisting of (1) sorting long-read libraries using variants that discriminate the two parental genomes of each cross and (2) performing separate de novo assembly on the sorted reads, similar to trio binning approaches (*Koren et al., 2018*). The proportion of sequenced bases successfully classified into subgenomes varied substantially depending on the cross (*Figure 2B*). CC crosses had the lowest classification rate (median: 70.8%) due to the scarcity of variants that discriminate *SpC* genomes from each other. BB crosses had a median classification rate of 84.5%, while the remainder (BC, BA, and BSc) reached a higher rate (median: 93.7%). Unclassified reads tended to be shorter than classified reads, but the difference was narrower for libraries with lower classification success (*Figure 2—figure supplement 1*). The scarcity of discriminating variants in low-divergence crosses is likely causing the classification of increasingly longer reads to fail. However, unclassified reads showed no overrepresentation in specific genomic regions (*Figure 2—figure supplement 2*) and their exclusion is thus unlikely to bias the resulting assemblies. The ratio of classified bases per subgenome was consistent with the corresponding ploidy levels: triploid BC lines had two copies of the *SpC* subgenome, while tetraploid lines had both *SpC* subgenomes duplicated (*Charron et al., 2019*; *Marsit et al., 2021*; *Figure 2B*).

Despite the variation in classification rates, the classified reads yielded highly contiguous subgenome-level assemblies, with a median of 22 contigs, a median N50 of 783 kb, and 95% of the assemblies having N50 values larger than 200 kb (*Figure 2C*, *Supplementary file 1a*). Only six assemblies were more fragmented (>100 contigs) and corresponded to the sorted libraries with the least sequenced bases (*Figure 2—figure supplement 3A*), which are expected to complicate the assembly. Thus, our approach yielded chromosome-level phased representations of the two subgenomes of each MA line.

We then integrated various data to infer the state of all loci containing Ty elements in the MA lines subgenomes in relation to the corresponding parental genome. First, we produced whole-genome assemblies of the 13 haploid parental strains from long reads. We annotated their Ty elements and used whole-genome alignments to determine the orthology of Ty loci between the parental strains of each cross (*Figure 3—figure supplement 1*). Only 10 pairs of loci were found to be orthologous in four of the lowest divergence crosses (CC1: 1; CC2: 5; CC3: 3; BB1: 1). We also annotated the Ty elements in the MA lines subgenome assemblies and defined the orthology between Ty loci by aligning subgenome assemblies against the corresponding parental assembly. We also computed the depth of coverage of sorted long-read libraries mapped on the corresponding parental genome assemblies to yield estimates of CN per genomic window. Finally, we used the ploidy level of each MA line subgenome as previously measured by flow cytometry and short-read sequencing (*Charron et al., 2019*; *Marsit et al., 2021*).

We integrated this data to classify individual Ty loci by type of SV using a hierarchy of binary rules (*Figure 3—figure supplement 2*). We note that the current methodology does not aim at providing an exhaustive quantification of all SVs in the MA lines, as previously done for some SV types (*Marsit et al., 2021*), but focuses solely on loci containing Ty elements. 89.8% of Ty loci were successfully

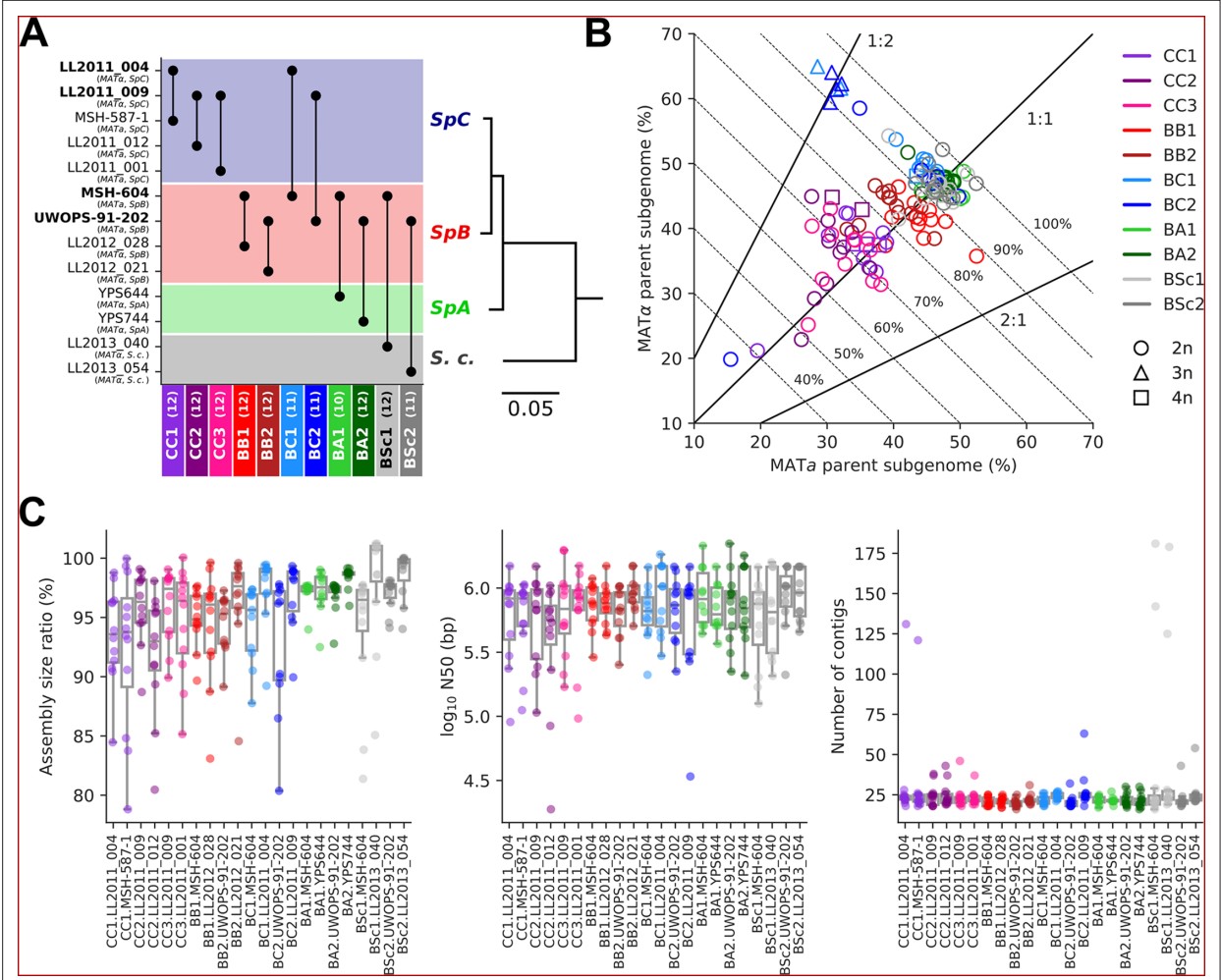

**Figure 2.** Long-read phasing and assembly yield subgenome-level representations of the genomes of hybrid yeast mutation accumulation (MA) lines. (**A**) Design of the MA crosses. Haploid derivatives of wild isolates used as parental strains for the MA crosses are shown on the Y-axis. Strains in bold are shared by multiple crosses. MA crosses are shown on the X-axis (left to right, from the least to the most divergent), with the number of lines randomly sampled for long-read sequencing shown in parentheses. The phylogenetic tree displays the evolutionary relatedness between wild populations of *S. paradoxus* and the sister species *S. cerevisiae*, based on genome-wide nucleotide variants (scale bar: substitutions per site) (*Hénault et al., 2022*). (**B**) Summary of the classification of long reads in two parental subgenomes. For each MA line library, the percentages of sequenced bases classified as the *MATa* and *MATα* parental subgenomes are shown on the X and Y axes, respectively. Dotted lines indicate global classification rates. Full lines indicate expected subgenome classification ratios of 1:1 for diploid (circles) or tetraploid (squares) lines, and 2:1 or 1:2 for triploid (triangles) lines. The ploidy level of each line was previously measured by DNA staining and flow cytometry (*Charron et al., 2019*; *Marsit et al., 2021*). (**C**) Summary statistics of the de novo subgenome-level assemblies of the MA lines. Assembly size ratio refers to the ratio of subgenome assembly size to the corresponding parental assembly size.

The online version of this article includes the following figure supplement(s) for figure 2:

**Figure supplement 1.** Read length distributions of classified and unclassified reads for a representative sample of four mutation accumulation (MA) lines long-read libraries.

**Figure supplement 2.** Genomic distribution of classified and unclassified reads.

**Figure supplement 3.** Subgenome-level assemblies of low contiguity.

classified, with the handful of low-contiguity assemblies contributing disproportionately to classification failure (*Figure 2—figure supplement 3B*). From the successfully classified loci, 79.0% had no change compared to the parental genomes. The remainder had an uneven distribution of SV types (*Figure 3A and B*). The largest class was polyploidy since many CC and BC lines have become tetraploid or triploid (*Charron et al., 2019*; *Marsit et al., 2021*). Triploidization was exclusive to six BC lines and was likely caused by the diploidization of *SpC* parental cells (creating mating-competent

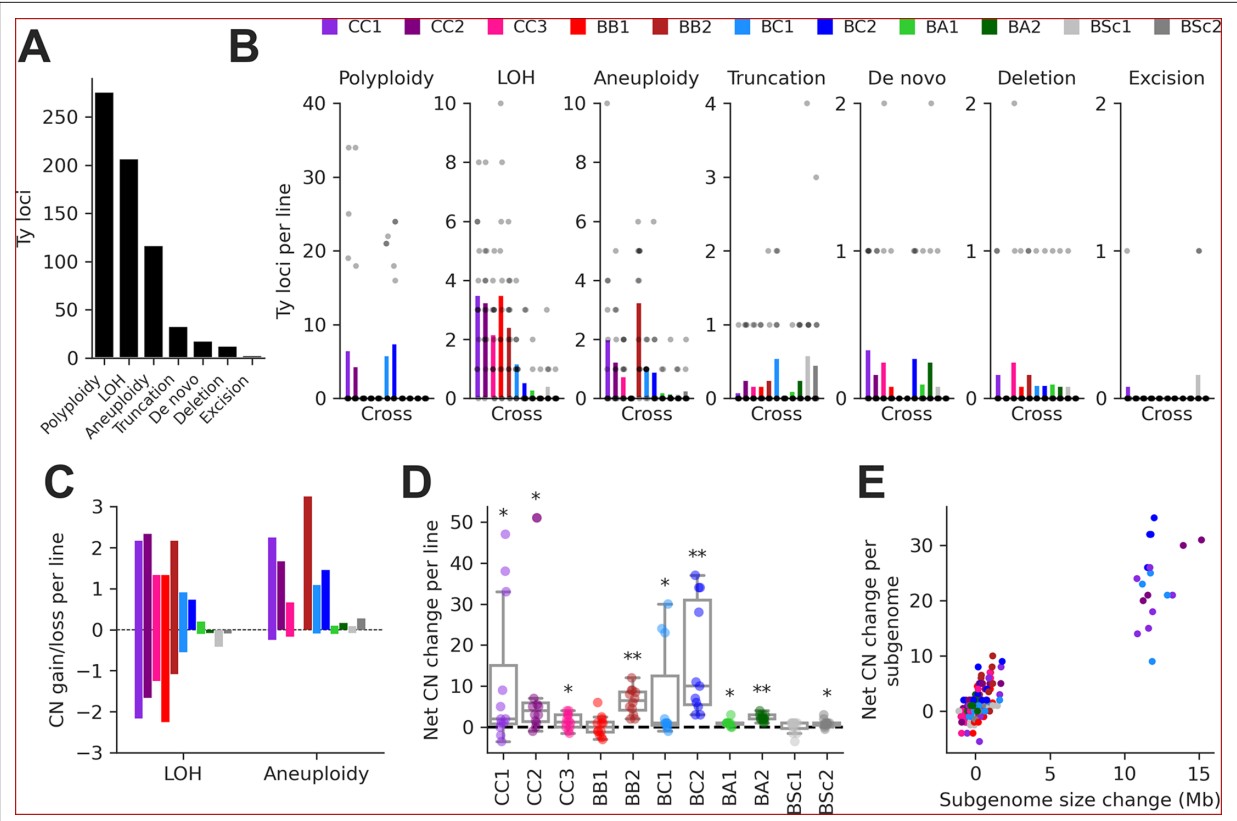

**Figure 3.** Structural variant (SV) types have unequal contributions to changes in the genomic distribution of Ty elements after evolution by mutation accumulation (MA). (**A**) Number of Ty loci affected by each SV type. (**B**) Number of Ty loci affected by SVs per line for each cross. Dots represent individual MA lines, and bars show the average per line. Polyploidy refers to increased ploidy level to triploidy or tetraploidy from the expected diploid initial state. Loss of heterozygosity (LOH) refers to the conversion of large chromosome segments (interstitial or terminal) from one parental genotype to the other. Aneuploidy refers to gains or losses of whole chromosomes. Truncation refers to partial deletion of a full-length Ty copy. De novo refers to the presence of a new full-length Ty copy not found in the corresponding parental background or any other MA line. Deletion refers to the local deletion of a full-length Ty copy. Excision refers to the replacement of a full-length Ty copy with a matching solo long terminal repeat (LTR), consistent with intra-element LTR-LTR recombination. (**C**) Average gain and loss of Ty copy number (CN) per line for each cross, for SV classes that can have bidirectional effects on Ty CN. (**D**) Distributions of net Ty CN change for individual MA lines. Symbols indicate false discovery rate (FDR)-corrected p-values for Wilcoxon signed-rank tests for deviations from zero (*p≤0.05, **p≤0.01). (**E**) Net Ty copy number change for individual MA line subgenomes as a function of subgenome size change.

The online version of this article includes the following figure supplement(s) for figure 3:

**Figure supplement 1.** Ty element annotations in the parental genomes.

**Figure supplement 2.** Hierarchy of binary rules used for the classification of Ty loci into structural variant (SV) types.

**Figure supplement 3.** Correlations between the estimations of full-length Ty copy numbers in the mutation accumulation (MA) lines genomes.

**Figure supplement 4.** Changes in Ty copy number (CN) are correlated with genome expansion.

pseudohaploids) prior to hybridization (*Charron et al., 2019*), meaning that triploidization occurred before MA strictly speaking. LOH and aneuploidy also accounted for large fractions of Ty loci affected by SVs. Notably, there was a substantial impact of LOH in the low-divergence CC crosses. Ty1 is the most abundant family in *SpC* genomes, and despite a relatively constant load, Ty1 copies mostly occupy non-orthologous loci (*Figure 3—figure supplement 1A*), providing the substrate for LOH to affect the landscape of Ty1 in CC crosses. While LOH had a bidirectional effect on Ty CNs, causing gains and losses in approximately equal proportions, aneuploidies had a strong bias toward gains (*Figure 3C*).

At the level of individual MA lines, the post-evolution Ty CNs previously quantified using relative depth of coverage of short-read libraries (*Hénault et al., 2020*) and the net CN changes determined here were globally correlated (mixed linear model with random intercepts for MA cross

× Ty family combinations, p-value = 3.01 × 10$^{-35}$). In many individual cases, significant positive correlations were observed (*Figure 3—figure supplement 3*), indicating that our current analyses recapitulate previous results. The distributions of net Ty CN change per MA line showed that most crosses had significant gains (*Figure 3D*), suggesting that Ty load can often increase as a result of random genetic drift. Some (but not all) of these crosses also exhibited significant increases in genome size after evolution (*Figure 3—figure supplement 4A*). The net Ty CN changes per MA line subgenome were globally correlated to the corresponding changes in subgenome size (*Figure 3E*). Even after excluding polyploid lines (which have the largest changes in both Ty CN and genome size), we found a significant relationship between the two variables (mixed linear model with random intercepts and slopes for MA crosses, p-value = 3.71 × 10$^{-9}$; *Figure 3—figure supplement 4B*), indicating that SVs affecting large portions of the genome have a substantial impact on the Ty landscape.

We identified 23 candidate de novo Ty insertions as loci occupied by one full-length Ty element that was absent from the corresponding parental genome or any other MA line. From these candidates, we used strict filtering criteria to narrow down confident loci compatible with retrotransposition. First, we required the presence of characteristic target site duplications (TSDs) of 5–6 nt (*Figure 4—figure supplement 1*), consistent with the staggered dsDNA cut catalyzed by the Ty1 integrase (*Wilhelm et al., 2005*). We further confirmed the uniqueness of Ty insertion junction sequences by inspecting whole-chromosome alignments comprising the MA line subgenomes and all parental genomes. We also ensured that all de novo TSD sequences were unique and absent from the parental full-length Ty loci. Eighteen de novo loci were compatible with all these criteria (*Figure 4*). The quantitative impact of de novo insertions on Ty load evolution is thus minor in comparison to other SV types, with 0.14 insertions per line on average. All de novo loci belonged to the Ty1 family of *S. paradoxus*. Of 11 crosses, 8 exhibited de novo Ty1 retrotransposition events, half of them being found in the CC crosses. Eight insertions were found at loci with parental Ty1 elements (full-length or solo-LTR) in close proximity, highlighting the importance of the de novo assembly approach to accurately resolve these complex loci. Two retrotransposition events in BC2 lines inserted into the UWOPS-91-202 *SpB* subgenome that is natively devoid of full-length elements, effectively re-colonizing a genome in which all Ty families had gone extinct.

We asked whether the de novo Ty1 insertions were associated with specific subfamilies from the parental genomes. We built a maximum likelihood phylogenetic tree comprising the de novo insertions and all parental Ty1 and Ty2 full-length sequences (*Figure 4—figure supplement 2*). This analysis revealed that de novo copies originated from major Ty1 clades populating *SpC, SpB,* and *SpA* genomes, indicating that Ty1 is globally active in *S. paradoxus* populations. However, this tree had generally poor branch support values from the bootstrap analysis. A phylogenetic network built from the same dataset showed substantial signal for reticulate evolution between Ty1 families from *S. cerevisiae* and *S. paradoxus SpA* (*Figure 4—figure supplement 3*), in agreement with previous studies that showed interspecific horizontal transfers (*Bleykasten-Grosshans et al., 2021*; *Czaja et al., 2020*).

Sequence resolution of de novo loci also enabled the characterization of insertions that disrupted parental genomic features. One de novo Ty1 element inserted into the left LTR of a parental full-length Ty1 copy in the MSH-587-1 subgenome of the MA line J27 (CC1 cross, *Figure 4—figure supplement 4A*), resulting in a nested insertion. Additionally, in a single case, a de novo Ty1 element inserted 37 bp downstream of the start codon of the host gene *AIM29* in the LL2011_009 subgenome of the MA line L41 (CC3 cross, *Figure 4—figure supplement 4B*), introducing an early stop codon that yields a presumably non-functional peptide of 36 amino acid residues in length.

The frequency of de novo Ty1 retrotransposition events per cross tends to support our previous conclusion that TE mobilization does not scale positively with parental divergence in *Saccharomyces* hybrids (*Hénault et al., 2020*). However, the number of de novo Ty1 insertions characterized by MA is low (at most four per cross), making it difficult to confidently test for differences in transposition rate between hybrid genotypes. In addition, multiple aspects of natural genetic variation were shown to impact the mobilization of Ty1 (*Czaja et al., 2020*; *Smukowski Heil et al., 2021*), suggesting that some of these aspects could also contribute to modulate transposition rate in *S. paradoxus* hybrids. Many of these aspects were not controlled for in the design of the MA experiment, which was primarily focused on generating increasingly high levels of parental divergence. Thus, it would be challenging to disentangle the effect of these factors with the limited sample sizes of our MA experiment.

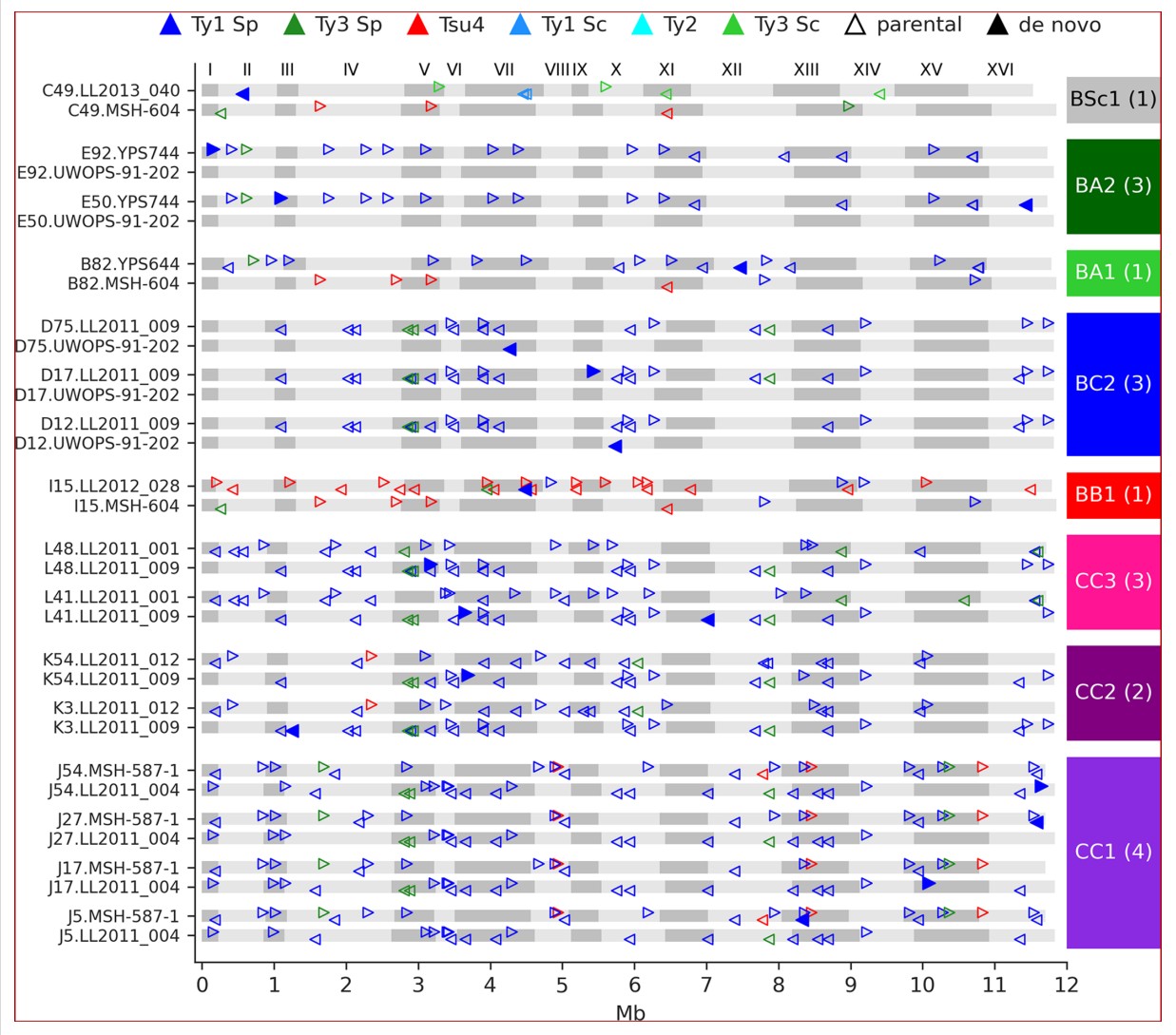

**Figure 4.** De novo retrotransposition events are rare in the mutation accumulation (MA) lines. Genome maps show the full-length Ty annotations from the subgenome assemblies of the MA lines with de novo insertions. Gray boxes represent chromosomes, which are numbered at the figure top. Empty triangles represent the parental annotations present at a copy number (CN) ≥1. Full triangles correspond to the confident de novo retrotransposition events. Triangle orientation indicates whether the annotation is on the plus (right pointing) or minus (left pointing) chromosome strand. The cross corresponding to each MA line is labeled at the right, with the total count of de novo retrotransposition events shown between parentheses. The average retrotransposition rates estimated from the de novo insertions (per line per generation per element) are the following: CC1, $1.0 \times 10^{-5}$; CC2, $4.9 \times 10^{-6}$; CC3, $7.6 \times 10^{-6}$; BB1, $1.5 \times 10^{-5}$; BC2, $1.7 \times 10^{-5}$; BA1, $6.5 \times 10^{-6}$; BA2, $2.2 \times 10^{-5}$; BSc1, $3.6 \times 10^{-5}$.

The online version of this article includes the following figure supplement(s) for figure 4:

**Figure supplement 1.** Target site duplication (TSD) sequences of the 18 confident Ty1 de novo retrotransposition events.

**Figure supplement 2.** Maximum likelihood phylogenetic tree comprising the full-length parental and de novo Ty1 and Ty2 elements.

**Figure supplement 3.** Phylogenetic network comprising the full-length parental and de novo Ty1 and Ty2 elements.

**Figure supplement 4.** De novo Ty1 transposition events disrupting parental genomic features.

We investigated how multiple aspects of genotype specificity impact transposition rate in vivo in *S. paradoxus* hybrids by adapting an existing genetic assay that measures the retrotransposition rate of a single tester Ty1 element from *S. cerevisiae* (*Curcio and Garfinkel, 1991*). This tool was notably used by *Smukowski Heil et al., 2021* to show that retrotransposition rates are not elevated in *S. cerevisiae × S. uvarum* hybrids. The assay relies on an antisense artificial intron (AI) that interrupts the coding sequence of a copy of the *HIS3* gene. This *HIS3*AI construct is itself inserted in antisense orientation in the 3' non-coding portion of the internal sequence of a plasmid-borne Ty1 element. The correct

splicing of the AI is only possible from the Ty1 mRNA, and the *HIS3* ORF can be restored once the spliced mRNA is reverse transcribed and integrated in the genome. When this construct is expressed in a strain auxotrophic for histidine, selection for histidine prototrophy in a fluctuation assay (*Luria and Delbrück, 1943*) enables the estimation of the transposition rate of the tester element. We adapted this assay to measure retrotransposition rates in multiple *S. paradoxus* backgrounds, and with two Ty1 sequence variants sampled from *SpB* and *SpC* genomes as tester elements (*Figure 5—figure supplement 1*, *Supplementary file 1b–f*).

The mechanism of negative copy-number-dependent self-regulation of retrotransposition (CNC) was characterized in the Ty1 family of *S. cerevisiae* (*Garfinkel et al., 2016*). This mechanism relies on the expression of an N-truncated variant of the Ty1 capsid/nucleocapsid Gag protein (p22) from two downstream alternative start codons (*Nishida et al., 2015*; *Saha et al., 2015*). The expression of p22 scales up with the CN of Ty1 elements that encode it (*Tucker et al., 2015*), which gradually interferes

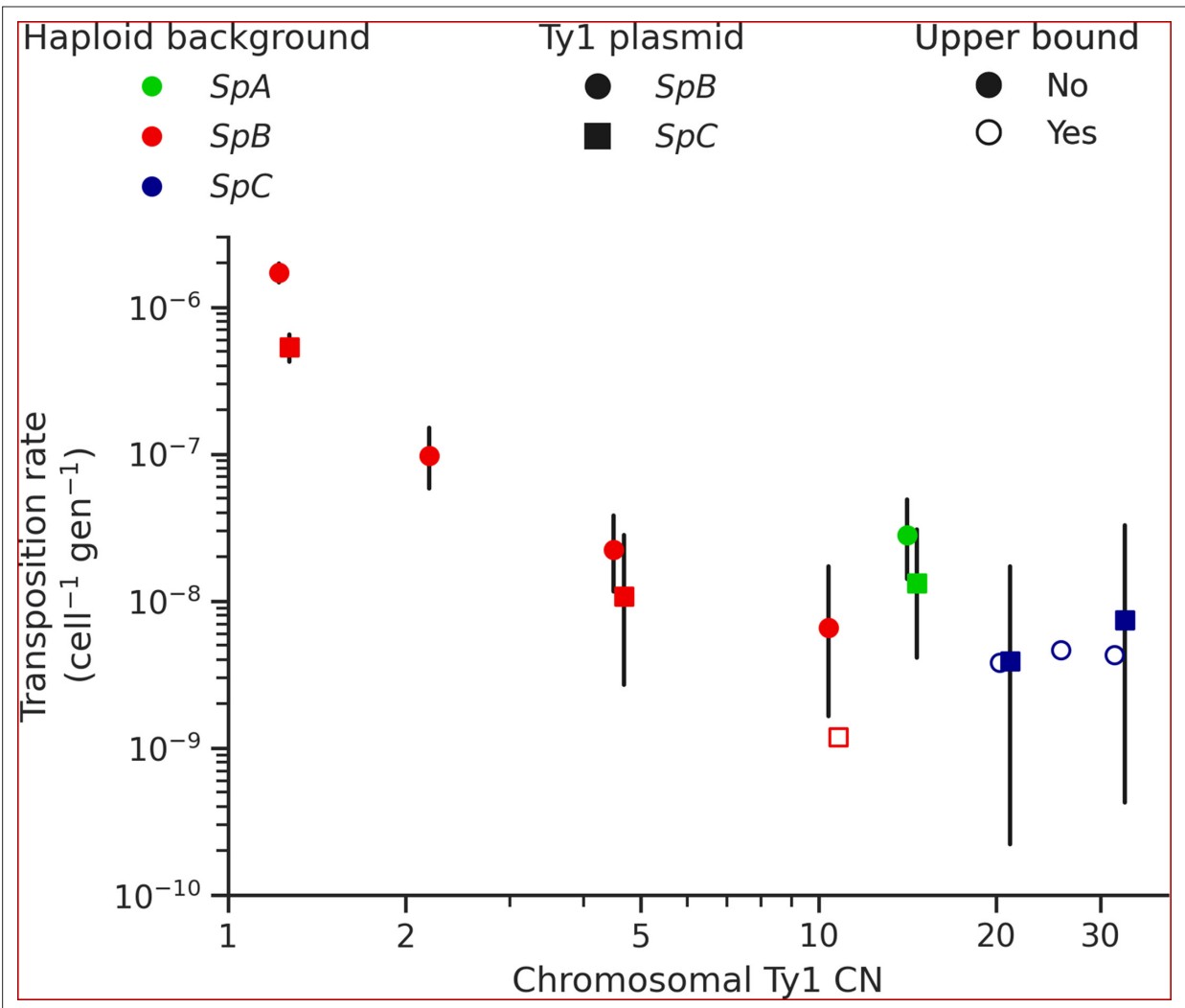

**Figure 5.** Ty1 transposition rate decreases as a function of chromosomal Ty1 copy number (CN) in *S. paradoxus*. Transposition rates were measured in eight haploid *S. paradoxus* backgrounds. Black bars indicate 95% confidence intervals. Empty symbols represent upper bound transposition rate estimates obtained by artificially inflating the mutant count by one when zero was observed. Marker shapes indicate the sequence variant of the tester Ty1 element, either *SpB* (circles) or *SpC* (squares). The *S. paradoxus* backgrounds and estimated Ty1 CNs are, from left to right, UWOPS-79-140 (*SpB*, 1.2); MSH-604 (*SpB*, 2.2); LL2012_021 (*SpB*, 4.6); LL2012_014 (*SpB*, 10.6); YPS744 (*SpA*, 14.4); LL2012_011 (*SpC*, 20.7); LL2011_012 (*SpC*, 26.3); and LL2011_009 (*SpC*, 32.4).

The online version of this article includes the following figure supplement(s) for figure 5:

**Figure supplement 1.** Methodology for the adaptation of the in vivo *S. cerevisiae* Ty1 retrotransposition assay to *S. paradoxus* Ty1 variants and genetic backgrounds.

with the assembly of the viral-like particles essential for Ty1 replication (*Cottee et al., 2021*; *Saha et al., 2015*). Thus, CNC yields a steep negative relationship between the retrotransposition rate measured with a tester element and the number of Ty1 copies in the genome (*Garfinkel et al., 2003*; *Tucker et al., 2015*). We previously observed a negative relationship between Ty1 load change after MA and the initial Ty1 load, and hypothesized that CNC-like regulation could be at play (*Hénault et al., 2020*). While the scarcity of Ty1 mobilization in MA lines now refutes this hypothesis, transposition rates in *S. paradoxus* could still be impacted by the initial load. We measured Ty1 transposition rates in eight different haploid backgrounds of *S. paradoxus* (one *SpA*, four *SpB*, and three *SpC*; *Supplementary file 1e and g*) that exhibit a wide variation in chromosomal Ty1 CN. The transposition rate in *SpB* backgrounds had a strong negative relationship with Ty1 CN (*Figure 5*). In an *SpB* background with a single Ty1 copy (UWOPS-79-140, estimated CN: 1.2), the rate reached $1.73 \times 10^{-6}$ cell$^{-1}$ gen$^{-1}$. In comparison, an *SpB* background harboring a single additional Ty1 copy (MSH-604, estimated CN: 2.2) had a transposition rate over 15 times lower ($9.77 \times 10^{-8}$ cell$^{-1}$ gen$^{-1}$). In *SpC* backgrounds, which all have higher Ty1 CNs (20.7–32.4), the transposition rate was below $1 \times 10^{-8}$ cell$^{-1}$ gen$^{-1}$.

In each background, we tested two sequence variants of the tester Ty1 element: one derived from a *SpB* genome (MSH-604) and one derived from a *SpC* genome (LL2011_012). We found that the *SpC* variant generally yielded lower transposition rates (*Figure 6A*). In the UWOPS-79-140 *SpB* background, the transposition rate of the *SpB* variant ($1.73 \times 10^{-6}$ cell$^{-1}$ gen$^{-1}$) was significantly higher than the *SpC* variant ($5.34 \times 10^{-7}$ cell$^{-1}$ gen$^{-1}$), with a difference of over threefold. We crossed three of the haploid backgrounds, one from each population (*SpB*: UWOPS-79-140; *SpC*: LL2011_012; and *SpA*: YPS744), to generate all the possible homozygous and heterozygous diploid backgrounds (*Supplementary file 1f*), and measured the transposition rate of both Ty1 variant in each background (*Figure 6B*, *Supplementary file 1h*). This experiment confirmed that transposition rates are lower with the *SpC* Ty1 variant, with the difference being significant in three different backgrounds (*SpA × SpB*, *SpB × SpB*, and *SpB × SpC*). Additionally, these measurements showed that transposition rates in heterozygous hybrids are intermediate between their respective homozygous parental backgrounds (*Figure 6—figure supplement 1*), providing further evidence for the robustness of Ty1 regulation in hybrids (*Hénault et al., 2020*; *Smukowski Heil et al., 2021*).

Like many other fungal species, mtDNA inheritance in *Saccharomyces* yeasts is biparental (*Wilson and Xu, 2012*), yielding a transient state of heteroplasmy that is resolved with the fixation of a single mtDNA haplotype by vegetative segregation (*Birky, 2001*). A recent study measured the rate of Ty1 transposition in *S. cerevisiae × S. uvarum* hybrids while controlling mtDNA inheritance and showed an effect of the parental mtDNA haplotype (*Smukowski Heil et al., 2021*). The transposition rate linked with the *S. uvarum* mtDNA haplotype was approximately one order of magnitude lower than the *S. cerevisiae* mtDNA haplotype. We investigated whether mtDNA inheritance also affected transposition rates at the intraspecific level in hybrids between *S. paradoxus* natural populations. For most of the diploid backgrounds described above, we tested reciprocal crosses with either parental haploid background being depleted of its mtDNA ($\rho^0$). As expected, the crosses with reciprocal $\rho^0$ parents in the homozygous *SpC* background yielded no significant difference in transposition rate (*Figure 7*). However, many significant differences were observed in heterozygous backgrounds. In both *SpA × SpC* and *SpB × SpC* hybrids with the *SpB* Ty1 tester variant, the transposition rate was significantly lower when the mtDNA was inherited from the *SpC* (*MATα*) parent (*Figure 7*). Although not statistically significant, the same trend was observed for the *SpC* Ty1 tester variant in the *SpB × SpC* hybrid. These results indicate a moderate but consistent effect of the *SpC* mtDNA in lowering the transposition rate of Ty1.

## Discussion

The repetitive nature of TEs makes the full sequence-level resolution of their genomic distribution notoriously difficult (*Goerner-Potvin and Bourque, 2018*). As such, studies investigating the TE reactivation hypothesis in hybrids have used a variety of methods for quantifying the activity of TEs at different levels, for instance, in terms of genomic abundance (*O'Neill et al., 1998*; *Ungerer et al., 2006*) or expression level (*Dion-Côté et al., 2014*; *Renaut et al., 2014*; *Drouin et al., 2021*). Not all methods enable the decomposition of TE load at the level of sequence-resolved insertion loci (*Tusso et al., 2022*), which is an especially challenging task in heterozygous genomes. A full mechanistic

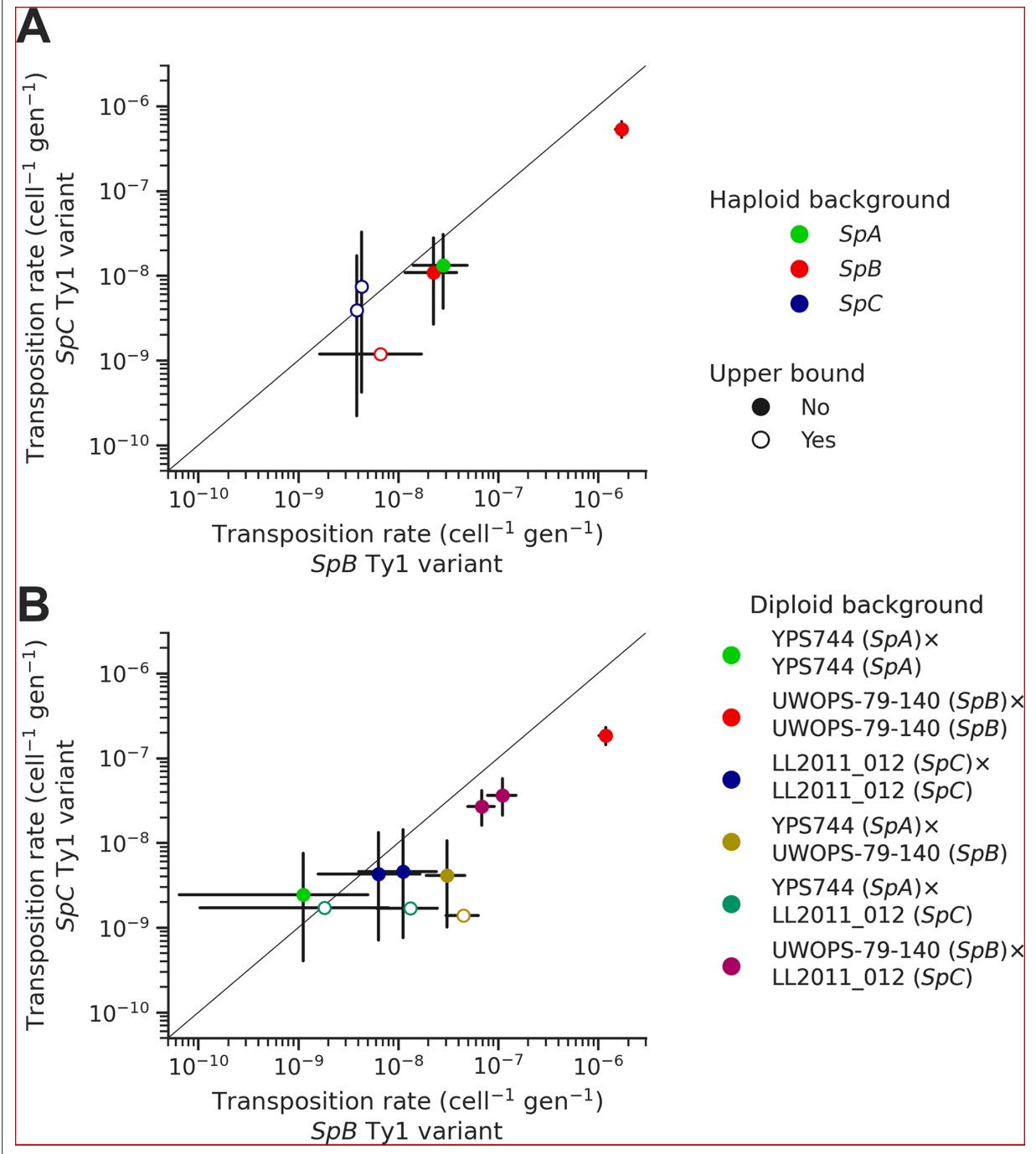

**Figure 6.** Natural sequence variants of the tester Ty1 element show distinct transposition rates. Transposition rates of the *SpB* (X-axis) and *SpC* (Y-axis) Ty1 variants are shown for haploid backgrounds (**A**) and diploid backgrounds (**B**). Black bars indicate 95% confidence intervals. Empty symbols represent upper bound transposition rate estimates obtained by artificially inflating the mutant count by one when zero was observed. The diagonal line indicates a 1:1 relation between transposition rates for the *SpB* and *SpC* Ty1 variants. The replicates of haploid backgrounds correspond to distinct strains from each population (see *Figure 5*), while the replicates for diploid backgrounds correspond to distinct mtDNA inheritance (see *Figure 7*).

The online version of this article includes the following figure supplement(s) for figure 6:

**Figure supplement 1.** Ty1 transposition rates in diploid homozygous backgrounds of natural populations of *S. paradoxus* and their hybrids.

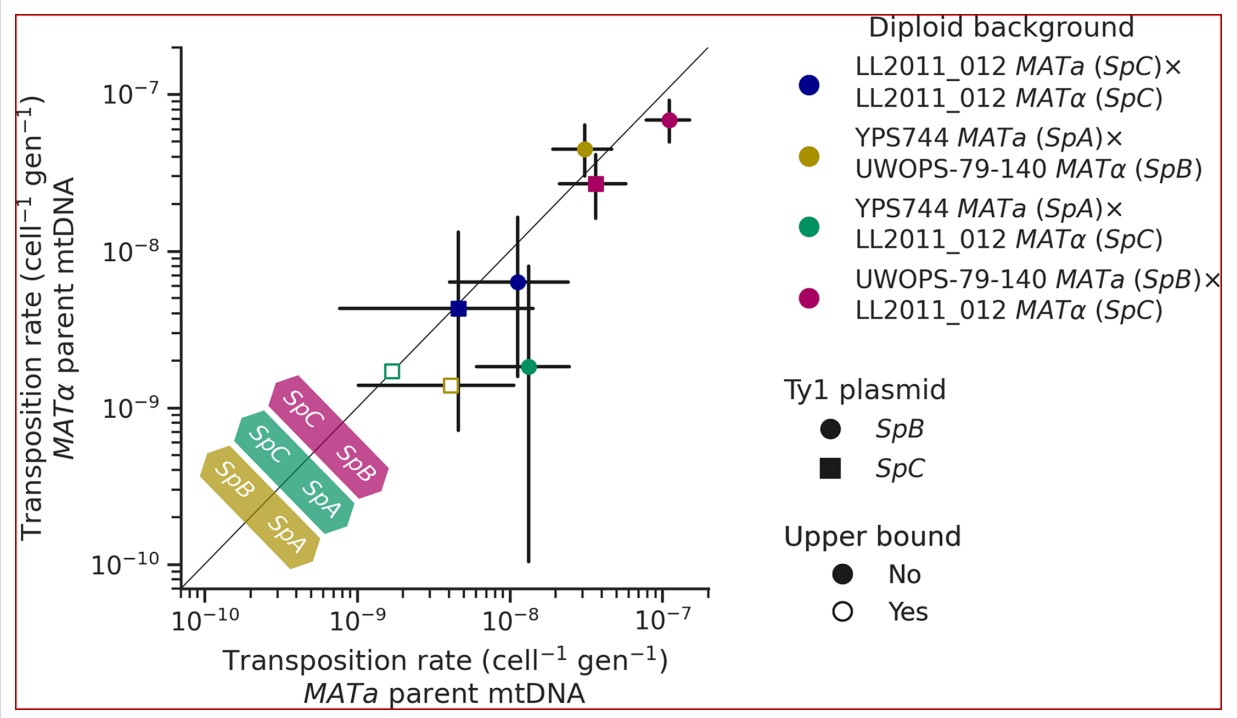

**Figure 7.** Uniparental mtDNA inheritance affects the transposition rate of Ty1 in heterozygous backgrounds. Transposition rates in diploid backgrounds as a function of whether the mtDNA is inherited from the *MAT*a parent (X-axis) or the *MAT*α (Y-axis) are shown. Black bars indicate 95% confidence intervals. Empty symbols represent upper bound transposition rate estimates obtained by artificially inflating the mutant count by one when zero was observed. Marker shapes indicate the sequence variant of the tester Ty1 element, either *SpB* (circles) or *SpC* (squares). The diagonal line indicates a 1:1 relation between transposition rates for reciprocal mtDNA inheritance. Colored arrows at the bottom left indicate the mtDNA haplotype that would correspond to a higher transposition rate on that side of the diagonal for each hybrid background.

understanding of how TE load evolves in hybrids requires the ability to characterize how TE copies are gained and lost, through the action of de novo transposition and other types of SVs.

Here, we used long-read sequencing to yield accurate representations of the genomes of yeast hybrids experimentally evolved under a near-neutral MA regime and investigated the mechanisms responsible for changes in the genomic landscape of Ty elements. We found that SV types like polyploidy, LOH, and aneuploidy had the largest impact on Ty CN variation. While de novo Ty1 insertions compatible with retrotransposition were found in most of the MA crosses, their contribution to CN changes was very modest compared to the transposition-unrelated SVs. The scarcity of de novo insertions in the MA lines provided insufficient resolution to dissect subtle transposition rate variation between hybrid genotypes. We thus complemented the genomic analysis of MA lines with an in vivo assay selecting for Ty1 mobilization and measured transposition rate in multiple genetic backgrounds of *S. paradoxus*. We found that initial Ty1 load, Ty1 sequence variation, and mtDNA inheritance modulate transposition rate. In addition, our results confirmed that unlike other yeasts (***Tusso et al., 2022***), Ty mobilization in *Saccharomyces* is not triggered by hybridization (***Hénault et al., 2020***; ***Smukowski Heil et al., 2021***) and this, along a broad gradient of parental evolutionary divergence.

Our experiments illustrate how under weakened natural selection efficiency TE load can increase in hybrid genomes by the action of transposition-unrelated SVs. This offers a nuanced perspective on the classical interpretation of the transposition-selection balance model (***Charlesworth et al., 1994***; ***Charlesworth and Langley, 1989***), in which increased TE load would be predominantly driven by the relaxation of purifying selection against TE insertions generated by de novo transposition. Our results suggest that SVs arising in the context of hybridization can act as a significant source of TE insertion polymorphisms which natural selection can purge more or less efficiently, depending on the population genetic context. This is closely related to the idea that sexual reproduction could favor the spread of TE families, contributing to their evolutionary success (***Hickey, 1982***; ***Zeyl et al., 1996***). Since the

insertion polymorphisms that contribute to increase TE load mostly originate from standing genetic variation, they could be less deleterious and thus harder for natural selection to purge efficiently.

Our in vivo transposition assays revealed sharp contrasts in the dynamics of Ty1 mobilization. Multiple aspects of the *SpC* genotypes are associated with lower mobilization rates. Ty1 transposition occurs at a very low rate in *SpC* backgrounds compared to other *S. paradoxus* populations, and using a *SpC* Ty1 sequence variant as a tester element showed a much lower transposition rate compared to a *SpB* Ty1 variant in matched backgrounds. Moreover, reciprocal crosses with controlled mtDNA inheritance suggested an effect of the *SpC* mtDNA haplotype in reducing Ty1 transposition rates. Despite the clear evidence for lower mobilization in vivo, half of the de novo transposition events in the MA lines genomes happened in intra-lineage *SpC* crosses. The *SpC* population harbors a low level of genetic diversity compared to *SpB* (*Eberlein et al., 2019*; *Leducq et al., 2016*) and has a fairly constant load of Ty1 (*Hénault et al., 2020*). Our new whole-genome assemblies revealed that this apparent CN stability hides major variation in terms of Ty1 insertion position, which explains how LOH events can drastically alter the Ty1 landscape in CC crosses.

We note that while the CC crosses tend to have the lowest retrotransposition rates as estimated from the de novo insertions (~$1 \times 10^{-5}$ per line per generation per element; *Figure 4*), these values are several orders of magnitude higher than the in vivo measures in *SpC* backgrounds. The discrepancy between these estimates could be due to uncharacterized biases inherent to each method. They could also be linked to differences between the parental genotypes used to generate the MA crosses and the fluctuation assays. One major difference is the use of *ade2* genotypes in the MA parents, a strategy that was initially adopted to provide a marker for the loss of mitochondrial respiration (*Joseph and Hall, 2004*; *Lynch et al., 2008*). It has been shown that the induction of adenine starvation through minimal adenine concentration in the medium and deletion of *ADE2*, which inactivates the adenine de novo biosynthesis pathway, increases Ty1 transcript levels (*Todeschini et al., 2005*), resulting in higher transposition rates. Rich complex medium like the one that was used for the MA experiment (YPD) can exhibit substantial variation in adenine concentration (*VanDusen et al., 1997*), and adenine can quickly become the limiting nutrient for *ade2* strains (*Kokina et al., 2014*). Thus, we cannot exclude that the choice of initial *ade2* genotypes could have inflated the transposition rates in the MA experiment.

The contrasted mobilization dynamics between natural *S. paradoxus* populations suggest that intraspecific variation affects not only the potency of Ty1 transposition, but also the strength of its regulation. *SpB* genomes generally have low Ty1 CNs, potent Ty1 sequence variants, and a steep negative relationship of transposition rate as a function of standing genomic Ty1 CN. This latter pattern is entirely consistent with the CNC regulation mechanism identified for Ty1 in *S. cerevisiae*, although the molecular demonstration of CNC function in our system remains to be done. On the other hand, *SpC* genomes have higher Ty1 loads, despite the observation that *SpC* Ty1 variants and mtDNAs are linked to markedly lower transposition rates. Our data suggest that transposition rates in *SpC* are much higher than what would be predicted by extrapolating the steep CNC-like negative relationship of *SpB* genomes (*Figure 5*). The less potent *SpC* Ty1 variant may have established a state that can maintain substantial polymorphism in the population, perhaps enabled by lower or non-functional CNC regulation. The first alternative start codon from which the p22 protein can be expressed in *S. cerevisiae* is conserved in both *SpB* and *SpC* Ty1 variants (M249, orthologous to M253 in the *S. cerevisiae* Gag protein sequence alignment from *Czaja et al., 2020*), suggesting that both *S. paradoxus* variants could express p22. Despite recent structural insights into the mechanism of p22-mediated CNC (*Cottee et al., 2021*), the molecular basis of the difference between CNC+ and CNC- p22 variants in *S. cerevisiae* remains unclear (*Czaja et al., 2020*). Variation outside of p22 is also likely to impact CNC. For instance, the expression of CNC components in *S. cerevisiae* is affected by the Mediator transcriptional regulator complex (*Salinero et al., 2018*), providing the host genome with a mechanism to modulate the action of CNC (*Curcio, 2019*). As such, intraspecific variation in Ty1 mobilization rate may stem from complex interactions between Ty1 potency, Ty1 self-regulation and host regulation.

In conclusion, our work provides a detailed account of the alteration of TE genomic landscapes in yeast hybrids evolved under relaxed natural selection efficiency through an accurate resolution of their subgenomes by phased assembly. We highlight that retrotransposition is only a minor contributor to changes in TE load compared to transposition-unrelated SVs. While in vivo transposition rate

assays revealed substantial variation that is underpinned by multiple aspects of genetic diversity at the intraspecific level, the potential for transposition rate in shaping the TE landscape in *Saccharomyces* hybrids is likely to be overpowered by other types of structural genomic variation.

# Materials and methods

## Experimental evolution by MA and long-read sequencing

The MA evolution experiments used in this study were described previously (*Charron et al., 2019*; *Hénault et al., 2020*). Briefly, crosses were made between 13 haploid derivatives of wild yeast isolates (11 *S. paradoxus* isolates and 2 *S. cerevisiae* isolates). A total of 11 crosses were designed to span a range of evolutionary divergence between the parental strains, from very low divergence at the intraspecific level to very high divergence at the interspecific level. Each cross was replicated between 48 and 96 times with independent matings. The resulting collection was evolved under an MA regime for approximately 770 mitotic generations by streaking for single colonies on YPD agar medium (*Supplementary file 1i*) at 3-day intervals. Glycerol stock archives were made of the entire collection at regular intervals, and a subset of 10–12 MA lines per cross were randomly selected for long-read sequencing. Long-read library preparation and sequencing were described in *Hénault et al., 2022*. Briefly, genomic DNA was prepared following a standard phenol-chloroform extraction and ethanol precipitation protocol. Native (PCR-free) genomic DNA libraries were prepared using the SQK-LSK109 kit (Oxford Nanopore Technologies [ONT]) and multiplexed with the EXP-NBD104 barcoding kit (ONT). Sequencing was performed on a MinION sequencer (MIN-101B, ONT) with FLOMIN106 (revC) flowcells. The sequencing was run using MinKNOW v3.3.2 (ONT) and the basecalling was run using guppy v3.0.3 (ONT). Demultiplexing was done using guppy_basecaller v3.1.5.

## Haploid genome assembly and annotation of the parental strains

De novo haploid genome assemblies for the 13 parental strains of the MA crosses were produced from basecalled long reads using wtdbg2 v2.5 (*Ruan and Li, 2020*) with parameters –p 0 –k 15 –AS 2 –s 0.05 –L 8000 –g 12m –X 120. Basecalled long reads were aligned to the draft assemblies using Minimap2 v2.17 (*Li, 2018a*) with preset map–ont. Draft assemblies were polished with the long-read alignments and the linked raw nanopore signal data using nanopolish v0.13.3 (*Loman et al., 2015*) with parameter ––min–candidate–frequency 0.1. Short-read libraries from the corresponding 13 parental strains were retrieved from NCBI (accession number PRJNA515073). Reads were trimmed using Trimmomatic v0.33 (*Bolger et al., 2014*) with parameters ILLUMINACLIP:{custom adapters file}:6:20:10 MINLEN:40. The short-read libraries were mapped on the signal-level polished draft assemblies using BWA-MEM v0.7.17 (*Li, 2013*) and SAMtools v1.15 (*Li et al., 2009*). The signal-level polished draft assemblies were polished with the short-read alignments using Pilon v1.22 (*Walker et al., 2014*).

Polished de novo genome assemblies were scaffolded at the chromosome level using the following procedure. Reference genomes for *S. paradoxus* strains LL2012_001 (*SpA*), MSH-604 (*SpB*), and LL2011_012 (*SpC*) (*Eberlein et al., 2019*) were scaffolded using RaGOO v1.1 (*Alonge et al., 2019*), using the genome of *S. cerevisiae* strain S288c (*Yue et al., 2017*) as a reference. For each of the MA parental strains, polished contigs were aligned against the scaffolded reference of the corresponding species or population using the nucmer (with preset ––mum) and show-coords (with parameters –r –d –T) tools from the MUMmer v3.23 suite (*Kurtz et al., 2004*). Contigs were sorted and merged into chromosomes using a custom Python v3.9.10 (*van Rossum and Drake, 2009*) script. We manually corrected for misassemblies that would yield dicentric or acentric chromosomes, or that were incompatible with karyotypes determined by CHEF-PFGE (*Charron et al., 2014b*). Chromosome-level assemblies were masked for rDNA repeats as follows. Repeats from the S288c reference genome (release R64-2-1, downloaded from https://www.yeastgenome.org) were extracted from chromosome XII (NC_001144, 451417-468931) and searched against the chromosome-level assemblies using BLASTN v2.11.0 (*Camacho et al., 2009*). Sequences corresponding to hits with alignment lengths over 200 bp were hard-masked using a custom Python v3.9.10 script.

Final assemblies were annotated for families of Ty LTR retrotransposons using RepeatMasker v4.1.2-p1 (*Smit et al., 2015*) with parameters –s –nolow –no_is –gccalc and a custom database of reference Ty sequences from *S. paradoxus* (*lib_Sp.fasta*) or *S. paradoxus* and *S. cerevisiae* combined (*lib.fasta*). RepeatMasker outputs were defragmented using REannotate version 26.11.2007 (*Pereira,*

*2008*) with parameters -s 15000 -d 10000. Final assemblies for the two parents of each MA cross were aligned to each other using the nucmer (with preset —mum) and show-coords (with parameters –r –d –T) tools from the MUMmer v3.23 suite. The orthology between the full-length Ty annotations for each cross was scored from visual inspection of the alignments.

## Sorting of reads per MA line subgenome

For each of the 11 crosses of the MA experiment, genomic variants that distinguish the two parental genomes were obtained from a variant call dataset previously generated (*Hénault et al., 2020*) from short reads mapped on the reference genome of *S. paradoxus SpB* strain MSH-604 (*Eberlein et al., 2019*). Informative variants were identified with bcftools v1.16 (*Li, 2011*) using a custom bash script. Long-read libraries were mapped against the MSH-604 genome (*Eberlein et al., 2019*) and a custom Python v3.9.10 script was used to classify individual reads as one parental genotype or the other. Each read had to have at least two informative variants. For a read to be classified as originating from the first parental subgenome, it had to be supported by a count of informative variants at least twice as high as the count of informative variants supporting the second parental subgenome. Lists of read IDs for reads successfully classified were used to split MA lines libraries using Seqtk v1.3-r106 (*Li, 2018b*).

## Subgenome-level assembly and annotation of MA line subgenomes

Basecalled long reads split by subgenome for each MA line were assembled separately using wtdbg2 v2.5 with parameters –p 0 –k 15 –AS 2 –s 0.05 –g 12m. Draft assemblies were polished with basecalled reads using medaka v1.4.4 (*Oxford Nanopore Technologies, 2021*) with model r941_min_high_g303. Polished assemblies were annotated for families of Ty LTR retrotransposons using the same methodology as for haploid parental genomes (see above). To enable the comparison of Ty annotations between the different assemblies, the following procedure was designed to establish a common coordinate system and define orthology relationships. Subgenome-level assemblies were aligned against the corresponding haploid parental genome using Minimap2 v2.20-r1061 with preset –x asm10. Ty annotation coordinates were exported into BED format using a custom Python v3.9.10 script and lifted over to haploid parental genome coordinates using paftools liftover v2.20-r1061 (*Li, 2018a*). The resulting annotation coordinates were merged in clusters (roughly corresponding to Ty orthogroups) in a custom Python v3.9.10 script using the DBSCAN algorithm from the scikit-learn v1.1.3 Python package (*Pedregosa et al., 2011*) with parameters eps = 500 min_samples = 1. Annotations within one cluster were required to share the same Ty family and strand orientation. Clusters were classified as complex or non-complex, depending on whether they contained more than one annotation in any genome.

## Classification of orthologous Ty clusters

For each cluster of orthologous Ty annotations in each MA line subgenome assembly, we assigned an SV class, if applicable. Unless otherwise stated, all the steps of the classification procedure were performed using custom Python v3.9.10 scripts.

Basecalled long reads split by subgenome were mapped against the corresponding parental assembly using Minimap2 v2.20-r1061 with preset map–ont, and alignments were filtered using SAMtools v1.16.1 to remove secondary alignments. Depth of coverage values from these alignments were extracted using SAMtools v1.16.1. Tracts of discrete relative depth of coverage along each genome were defined as follows. Assemblies were subdivided into 30-kb-wide non-overlapping windows, and the median depth of coverage value of each window was divided by the genome-wide median. Adjacent windows having the same value were merged to generate tracts that reflect major CN changes along the genome (i.e., polyploidy, aneuploidies, and LOH events).

In addition to the discrete estimates of relative depth of coverage along each subgenome, the existence of a subgenome-level assembly was inferred for each parental Ty cluster using a reverse liftover procedure as follows. The parental genomes were aligned against the corresponding subgenome-level assemblies using Minimap2 v2.20-r1061 with preset –x asm10. For each Ty cluster, the two 500-bp-wide windows immediately flanking the annotation were exported into BED format and lifted over to subgenome-level assembly coordinates using paftools liftover v2.20-r1061. For the downstream steps, the clusters having inconsistent depth of coverage-based CN estimates and assembly

alignment-based presence (i.e., having a CN of zero and assembly presence, or a CN greater than zero and assembly absence) were excluded.

These data were organized as Boolean variables as follows. At the level of the individual orthologous cluster, the variables included were whether the cluster is complex (comprising more than a single annotation per genome) and whether the cluster contains an annotation in the corresponding parental genome. At the level of the individual orthologous cluster in individual MA line subgenomes, the variables included were whether the cluster contains an annotation, whether there is the presence of an assembly based on assembly alignment, whether the content of the cluster is the same as the parent, and whether the CN of the cluster is the same as the parent. The CN of the cluster and the types of annotations in the cluster are non-Boolean variables that were also included at this level.

Using these variables, a hierarchy of binary rules (*Figure 3—figure supplement 2*) was used to attribute one of the following SV classes for each orthologous cluster in each MA line subgenome. The majority of clusters were identical to the corresponding cluster in the parental genome and were labeled 'No change'. The clusters which had identical contents to the parental genome, but in a different CN, were labeled 'Polyploidy/aneuploidy/LOH'. The clusters which had a full-length annotation in the parent and a truncated annotation in the MA line subgenome were labeled 'Truncation'. The clusters which had a full-length annotation in the parent and a solo LTR in the MA line subgenome were labeled 'Excision'. The clusters which had a unique additional full-length annotation in the MA line subgenome compared to the parent were labeled 'De novo'. The clusters which had an annotation in the parental genome, but no annotation in the MA line subgenome despite having support for MA line subgenome assembly presence, were labeled 'Deletion'.

We manually classified loci from the class 'Polyploidy/aneuploidy/LOH' using the mappings of long-read libraries against parental assemblies. Genome-wide relative depth of coverage visualizations were generated by binning values in non-overlapping windows of 10 kb and normalizing the median of each bin by the genome-wide median value. Profiles were individually examined to determine whether CN changes were due to polyploidy, aneuploidy or large tracts of LOH.

Genome size change after evolution for each MA line was computed from the 10 kb bins described above. Bins with extreme values (below the 1st percentile or above the 99th percentile) were excluded. The difference in total depth for one subgenome and the corresponding haploid parental genome was computed to yield the change in genome size in Mb.

## Identification of de novo transposition events

From the classification produced by the decision tree, the loci identified as 'De novo' were curated to identify confident loci corresponding to retrotransposition events. We extracted the sequences immediately flanking the insertion loci and looked for short repeated motifs in direct orientation, corresponding to TSDs. TSDs of five or six nucleotides were found, and loci with no identifiable TSD were discarded. We produced chromosome alignments comprising the MA line subgenome(s) with de novo loci and all parental strains using Mauve v2015-02-26 (*Darling et al., 2010*) with the progressiveMauve algorithm and default parameters. Alignments were inspected to confirm that the junctions of de novo loci were unique to the corresponding MA line subgenome. Finally, we compared the TSD motifs of candidate de novo retrotransposition to TSD motifs of all parental full-length copies to filter out non-unique ones. The remaining loci (all from the Ty1 family) were classified as de novo retrotransposition events.

A phylogenetic analysis of the de novo Ty1 insertions was performed by extracting the sequences of de novo loci and of all full-length Ty1 and Ty2 parental elements. We added the sequences of representative Ty1 and Ty2 elements identified by *Bleykasten-Grosshans et al., 2021* as references. We included the 50 bp flanking sequences in 5' and 3' of all extracted sequences and ran a multiple sequence alignment using MUSCLE v3.8.31 (*Edgar, 2004*). The alignment was trimmed manually using MEGA v11.0.11 (*Tamura et al., 2021*). A maximum likelihood phylogenetic tree was computed on the trimmed alignment using RAxML-NG v1.1.0 (*Kozlov et al., 2019*) with the GTR+G model initiated with 10 parsimony trees and 200 bootstraps. The phylogenetic tree was visualized using FigTree v1.4.4 (*Rambaut, 2018*). Additionally, we built a phylogenetic network from the same trimmed alignment using SplitsTree v4.18.3 (*Huson and Bryant, 2006*; *Huson and Bryant, 2022*) with default parameters.

## Plasmids used and constructed in this study

The plasmids used and constructed in this study are listed in *Supplementary file 1b*. Oligonucleotide sequences are detailed in *Supplementary file 1c*. PCR reactions and cycles are detailed in *Supplementary file 1d*. Media recipes are detailed in *Supplementary file 1i*. All the PCRs used for cloning were performed using KAPA HiFi HotStart DNA polymerase (Roche). All the PCRs used for cloning confirmations were performed using Taq DNA Polymerase (BioShop). Plasmid minipreps were performed on overnight cultures in 5 mL 2YT+Amp or 2YT+Kan medium, depending on the plasmid, using the Presto Mini Plasmid Kit (GeneAid).

Gibson cloning reactions were performed as follows. A Gibson assembly master mix was prepared with the following components (for 1.2 mL): 360 µL of 5× ISO buffer (500 mM Tris–Cl, 50 mM MgCl₂, 4 mM dNTPs mix, 50 mM DTT, 250 mg mL⁻¹ PEG 8000, 5 mM NAD⁺), 0.64 µL 10 U µL⁻¹ T5 exonuclease (New England Biolabs), 20 µL of 2 U µL⁻¹ Phusion High-Fidelity DNA Polymerase (New England Biolabs), 160 µL of 40 U µL⁻¹ Taq DNA ligase (New England Biolabs), and 700 µL of PCR-grade H₂O. Gibson reactions were assembled by adding all the components to 7.5 µL of Gibson master mix, and completing the volume to 10 µL with PCR-grade H₂O. Gibson reactions were incubated at 50°C for 60 min.

Competent cells of *Escherichia coli* were prepared as detailed in *Green and Rogers, 2013*. *E. coli* transformations were performed by combining 100 µL of competent cells with 5 µL of mutagenesis or Gibson assembly reaction and incubating the mixture on ice for 15 min. Heat shock was done at 42°C for 1 min, followed by incubation on ice for 5 min. Then, 900 µL of 2YT medium was added and the mixture was incubated for 45 min at 37°C with agitation. Cells were plated on 2YT+Amp agar or 2YT+Kan agar medium, depending on the plasmid, and incubated at 37°C for 24 hr.

## Construction of Ty1 retrotransposition marker plasmids

Retrotransposition rate measurement assays were developed based on the plasmid-borne assay designed by *Curcio and Garfinkel, 1991* with several modifications. First, we replaced the *S. cerevisiae* Ty1 sequence with variants sampled from wild *S. paradoxus* genomes. To measure transposition rates at a transcription rate closer to the natural genomic context and prevent potential biases arising from galactose induction in multiple wild genetic backgrounds, we opted for the native promoter of Ty1 variants instead of the *GAL1* promoter. Finally, we opted for a low CN centromeric plasmid backbone instead of a high CN plasmid backbone in order to increase the mitotic stability and reduce the CN variance of the system.

Plasmid pRS31N was constructed as follows. Using a pFA6-natNT2 (*Janke et al., 2004*) miniprep as a template, we amplified the natNT2 nourseothricin (Nat) resistance cassette. Using a pRS316 (*Sikorski and Hieter, 1989*) miniprep as a template, we amplified the backbone excluding the *URA3* selection marker. A Gibson assembly mix was prepared by adding 1 µL of natNT2 cassette and 1 µL of backbone to the Gibson master mix. After incubation for 60 min at 50°C, the Gibson reaction was digested with DpnI (New England Biolabs). Competent cells of *E. coli* MC1061 (*Casadaban and Cohen, 1980*) were transformed with 5 µL of the Gibson reaction. Cloning was confirmed by PCR by testing the 5' junction. A miniprep was migrated by agarose gel electrophoresis and transformed into yeast, selecting on both YPD+Nat agar medium and SC-ura agar medium.

Plasmids pRS31N-Ty1-*SpB*-*HIS3*AI and pRS31N-Ty1-*SpC*-*HIS3*AI were constructed as follows. The plasmid pGTy1-H3*HIS3*AI was a gift from David Garfinkel (Addgene plasmid # 62228; RRID:Addgene_62228; *Curcio and Garfinkel, 1991*). Using a pGTy1-H3m*HIS3*AI miniprep as a template, we amplified the retrotransposition indicator gene (RIG) *HIS3*AI. We treated the *HIS3*AI RIG amplicon with DpnI followed by purification using solid-phase reversible immobilization (SPRI) beads. We digested the plasmid pRS31N with SacII (New England Biolabs), treated the digestion with calf intestinal alkaline phosphatase (CIP, New England Biolabs), and purified the digested plasmid using SPRI beads. We extracted genomic DNA from two *S. paradoxus* wild diploid strains (MSH-604, *SpB*; and LL2011_012, *SpC*) by following a standard phenol-chloroform extraction protocol on overnight cultures in 5 mL YPD medium incubated at 30°C with agitation. A genomic region of ~10 kb containing a single full-length Ty1 element was amplified from the MSH-604 (chrXV) and LL2011_012 (chrVII) genomic DNA extractions using unique primers. The full-length element chosen from the LL2011_012 genome was identical to five other full-length Ty1 elements in the same genome, and one of the two full-length Ty1 elements in MSH-604 was chosen at random. From each genomic region amplicon, the full-length

Ty1 element was amplified in two segments: a first segment spanning the left LTR and complete internal ORF (Ty1 5' segment), and a second segment spanning the right LTR (Ty1 3' segment). The Ty1 segment amplicons were purified with SPRI beads. A first Gibson assembly reaction was performed by combining 40 fmol each of the digested pRS31N backbone, the Ty1 5' segment amplicon, and the *HIS3*AI RIG amplicon. Competent cells of *E. coli* MC1061 were transformed with 5 μL of the Gibson assembly reaction. Cloning was confirmed by PCR by testing the 5' and 3' junctions. We prepared a miniprep of the first Gibson assembly and validated its restriction profile with EcoRV (New England Biolabs). The miniprep of the first Gibson assembly was digested with NotI (New England Biolabs), and the digestion was treated with CIP and purified using SPRI beads. A second Gibson assembly reaction was performed by combining 40 fmol each of the first Gibson assembly digestion and the Ty1 3' segment amplicon. Cloning was confirmed by PCR by testing the 5' and 3' junctions. We prepared a miniprep of the second Gibson assembly and validated its restriction profile with EcoRV. We confirmed the final clones with long-read whole-plasmid sequencing at Plasmidsaurus (http://www.plasmidsaurus.com).

## Yeast strains used and constructed in this study

*S. paradoxus* strains used and constructed in this study are listed in *Supplementary file 1e*. Oligonucleotide sequences are detailed in *Supplementary file 1c*. PCR reactions and cycles are detailed in *Supplementary file 1d*. Media recipes are detailed in *Supplementary file 1i*. All the PCRs for plasmid mutagenesis and cassette amplification were performed using the KAPA HiFi HotStart DNA polymerase. All the PCRs used for confirmations or diagnostics were performed using the Taq DNA Polymerase.

 *S. paradoxus* competent cells were prepared from 25 mL cultures in YPD medium incubated at 30°C with agitation until they reached an optical density at 600 nm ($OD_{600}$ $mL^{-1}$) of 0.5–0.7. Cultures were centrifuged at $500 \times g$ for 5 min. Cells were washed once in 5 mL in $H_2O$ and once in 5 mL SORB solution (1 M sorbitol, 100 mM lithium acetate, 10 mM Tris–Cl, 1 mM EDTA). Cells were resuspended in 180 μL of SORB solution and 20 μL of salmon sperm DNA that was previously boiled for 5 min and cooled on ice. Competent cells were stored at –70°C until use.

 *S. paradoxus* transformations were performed by combining 20 μL of competent cells, 250 ng of plasmid or 8 μL cassette amplicon, and 100 μL of Plate Mixture solution (40% PEG 3350, 100 mM lithium acetate, 10 mM Tris–Cl, 1 mM EDTA). The mixture was incubated at room temperature for 30 min. Then, 7.5 μL of DMSO were added and a heat shock was performed at 37°C for 30 min. Cells were centrifuged at 2000 rpm for 3 min and resuspended in 100 μL of YPD medium for 2 hr in the case of plasmid transformations, or 5 hr in all other cases.

## Generation of histidine auxotroph strains

Short placeholder sequences (hereafter referred to as stuffers) (*Dionne et al., 2021*) to be used as donor DNA in CRISPR-Cas9-mediated deletion of *HIS3* (YOR202W) were amplified using the following reactions: 2 μL of Taq 10× Buffer (BioShop), 1.2 μL of 25 mM $MgCl_2$, 0.4 μL of 10 mM dNTPs mix, 0.4 μL of 10 μM 5' primer, 0.4 μL of 10 μM 3' primer, 2 μL of 0.01 μM template primer, 0.12 μL of Taq DNA Polymerase, and 13.5 μL of PCR-grade $H_2O$. The following PCR cycle was run: 3 min at 94°C; 35 cycles of 30 s at 94°C, 30 s at 68°C, and 15 s at 72°C; 25 s at 72°C; hold at 10°C.

 A plasmid encoding CRISPR-Cas9 and a guide RNA (gRNA) sequence targeting the *HIS3* coding sequence in *S. paradoxus* was designed as follows. The plasmid pCAS was a gift from Jamie Cate (Addgene plasmid # 60847; RRID:Addgene_60847; *Ryan et al., 2014*). pCAS was used as a template to mutagenize the gRNA sequence to target the coding sequence of the *PEX11* gene (YOL147C). Mutagenesis PCRs were treated with DpnI and transformed in *E. coli* MC1061. The mutagenesis was confirmed by Sanger sequencing of the gRNA locus from a miniprep. pCAS-*PEX11* was used as a template to mutagenize the gRNA sequence to target the coding sequence of *HIS3*. Mutagenesis PCRs were treated with DpnI and transformed in *E. coli* MC1061. The gRNA locus was amplified for confirmation, and the mutagenesis was confirmed by digesting the amplicon with HinfI (New England Biolabs) (the *PEX11* gRNA sequence contains a HinfI restriction site, while the *HIS3* gRNA doesn't) and by Sanger sequencing.

 CRISPR-Cas9-mediated deletion of *HIS3* was done following the method of *Ryan et al., 2016* with pCAS-*HIS3* and the corresponding stuffer amplicon in eight different haploid *S. paradoxus* strains

derived from wild isolates (*Charron et al., 2014b*; *Leducq et al., 2016*). Gene deletion was confirmed by PCR and growth inability on SC-his agar medium.

To enable crosses among three of the *his3*::stuffer strains, one per population (*SpA*: YPS744, *SpB*: UWOPS-79-140, *SpC*: LL2011_012), the hphMX hygromycin B (HygB) resistance cassette at the *HO* locus was replaced by the kanMX geneticin (G418) resistance cassette, and the mating types were switched.

The plasmid pUG6 (*Güldener et al., 1996*) was obtained from Euroscarf (http://www.euroscarf.de). pUG6 was used as a template to amplify the kanMX cassette with flanking homology to the HO locus. Competent cells were transformed with the cassette and plated on YPD+G418 agar medium. Cassette switch was confirmed by PCR.

The plasmid pHS3 was a gift from John McCusker (Addgene plasmid # 81038; RRID:Addgene_81038). Competent cells from the mating-type switched strains were transformed with pHS3 and plated on YPD+G418+Nat agar medium. Colonies were grown in 3 mL YPD medium at room temperature with agitation for 2 d to allow for pHS3 segregation, diluting cultures ~1000× after 24 hr. Cultures were streaked on YPD plates and incubated at room temperature for 5 d. Single colonies were checked for the loss of pHS3 by making patches on YPD+Nat agar medium and for autodiploidization by PCR (*Huxley et al., 1990*). Strains were sporulated as described in *Plante and Landry, 2020*, and tetrads were dissected on YPD agar medium using a SporePlay microscope (Singer Instruments). Dissection plates were incubated at room temperature for 5 d. The mating types of spores were confirmed by PCR (*Huxley et al., 1990*).

To enable crosses with reciprocal mtDNA inheritance, we generated strain derivatives devoid of mtDNA ($\rho^0$) as described in *Fox et al., 1991*. The loss of respiration ability was confirmed by spotting the strains on YPEG agar medium.

## Crosses with reciprocal mtDNA inheritance

Histidine auxotroph strains and their *HO* cassette-switch, mating type-switch, and $\rho^0$ derivatives were crossed as detailed in *Supplementary file 1f*. For each parental strain, an overnight preculture was grown in 1 mL of YPD medium at room temperature. Then, 100 µL of each preculture were combined in 1 mL YPD medium and incubated at room temperature without agitation for 4 hr. Also, 10 µL of mating cultures were streaked on YPD+G418+HygB agar medium and incubated for 3 d at room temperature. Two single colonies per streaking were used to inoculate 1 mL of YPD medium, and the strains were tested for respiratory proficiency by spotting on YPEG agar medium. The strains were also tested for diploidy by cellular DNA staining and fluorescence measurement by flow cytometry using a Guava easyCyte 8HT flow cytometer (EMD Millipore) as described in *Marsit et al., 2021*.

## Measurement of retrotransposition rates

Fluctuation assays were performed to measure retrotransposition rates. Competent cells of the corresponding haploid strains and diploid crosses were transformed with minipreps of either pRS31N-Ty1-*SpB*-*HIS3*AI or pRS31N-Ty1-*SpC*-*HIS3*AI, and plated on YPD+Nat agar medium. Transformation plates were incubated at 30°C for 3 d. From the transformation plates, single colonies were picked and resuspended in 10 mL of YPD medium. Cell densities were estimated by measuring the $OD_{600}$ of the resuspensions, and dilutions at $2 \times 10^{-3}$ $OD_{600}$ mL$^{-1}$ were prepared in YPD medium. 96-well plates were filled with 225 µL of YPD+Nat 1.1× medium. Then, 25 µL of cell dilutions were added to each well for a final concentration of $2 \times 10^{-4}$ $OD_{600}$ mL$^{-1}$ (~500 cells per well). Each combination of background and Ty1 variant was replicated in 48 independent cultures (half a plate). Plates were covered with breathable rayon films and incubated at 20°C for 3 d. Final cultures were centrifuged at 500 × *g* for 5 min and resuspended in 200 µL of H$_2$O. The entire resuspensions were spotted in groups of 7 or 9 on SC-his agar plates that were previously dried at 37°C for 3 d. Plates were incubated at room temperature and His$^+$ colonies were counted after 5 d.

For each haploid background, 42 cultures were spotted on SC-his agar plates and six were pooled. Serial dilutions were prepared by diluting pools twice 200× and 100 µL of final dilutions were plated on YPD agar plates. Plates were incubated at room temperature for 3 d, and colonies were counted to estimate final population sizes. Colony counts are detailed in *Supplementary file 1g*. For each diploid background, 36 cultures were spotted on SC-his agar plates and 12 were pooled. Serial dilutions were prepared by diluting pools twice 12×, and cell densities of the final dilutions were measured by flow

cytometry using a Guava easyCyte 8HT flow cytometer to estimate final population sizes. Colony counts are detailed in *Supplementary file 1h*.

The number of mutations (retrotransposition events) in each culture and associated 95% confidence intervals were estimated using the newtonLD and confintLD functions from the Python implementation of the R package rSalvador (*Zheng, 2017*; *Zheng, 2020*). Retrotransposition rates were computed by dividing mutation count estimates by the estimates of final population sizes.

## Acknowledgements

We thank Alexandre Dubé for helpful comments on the manuscript and contributions to plasmid constructions. We thank Anna Fijarczyk and Hélène Martin for contributions to sequencing data preprocessing. We thank the members of the Landry laboratory for helpful discussions on the project. This project was supported by funding to CRL from an NSERC discovery grant (RGPIN-2020-04844) and an FRQNT team grant (2019-PR-254415), and an NSERC Alexander Graham Bell doctoral scholarship to MH. CRL holds the Canada Research Chair in Cellular Systems and Synthetic Biology.

## Additional information

### Competing interests

Christian R Landry: Senior editor, *eLife*. The other authors declare that no competing interests exist.

### Funding

| Funder | Grant reference number | Author |
| --- | --- | --- |
| Natural Sciences and Engineering Research Council of Canada | RGPIN-2020-04844 | Christian R Landry |
| Fonds de recherche du Québec – Nature et technologies | 2019-PR-254415 | Christian R Landry |

The funders had no role in study design, data collection and interpretation, or the decision to submit the work for publication.

### Author contributions

Mathieu Hénault, Conceptualization, Resources, Data curation, Formal analysis, Funding acquisition, Investigation, Visualization, Methodology, Writing – original draft, Writing – review and editing; Souhir Marsit, Guillaume Charron, Conceptualization, Resources, Investigation, Methodology; Christian R Landry, Conceptualization, Resources, Supervision, Funding acquisition, Validation, Methodology, Project administration, Writing – review and editing

### Author ORCIDs

Mathieu Hénault ⓘ https://orcid.org/0000-0003-0760-7545
Christian R Landry ⓘ http://orcid.org/0000-0003-3028-6866

Reviewer #1 (Public Review): https://doi.org/10.7554/eLife.89277.3.sa1
Reviewer #2 (Public Review): https://doi.org/10.7554/eLife.89277.3.sa2
Author Response https://doi.org/10.7554/eLife.89277.3.sa3

## Additional files

### Supplementary files

• Supplementary file 1. Supplementary tables. (a) Subgenome-level assemblies of MA lines. (b) Plasmids used an constructed in this study. (c) Oligonucleotides sequences used in this study. (d) PCR reactions and cycles used in this study. (e) Strains used and constructed for the retrotransposition rate measurement assays. (f) Crosses generated for the retrotransposition rate

measurement assays. (g) Raw data of the fluctuation assays on haploid backgrounds. (h) Raw data of the fluctuation assays on diploid backgrounds. (i) Media recipes used in this study.

- MDAR checklist

### Data availability

Nanopore long-read sequencing data (basecalled reads) and parental genome assemblies are available at NCBI under accession number PRJNA828354. Illumina short-read sequencing data of the MA lines is available at NCBI under accession number PRJNA515073. Custom scripts, custom databases of reference Ty sequences and subgenome-level assemblies of MA lines used in this study are available at GitHub (copy archived at *Landrylab, 2023*).

The following datasets were generated:

| Author(s) | Year | Dataset title | Dataset URL | Database and Identifier |
|---|---|---|---|---|
| Henault M, Marsit S, Charron G, Landry CR | 2022 | ONT whole genome sequencing from MA experiment on *S. paradoxus* and *S. cerevisiae* wild yeast hybrids | https://www.ncbi.nlm.nih.gov/bioproject/?term=PRJNA828354 | NCBI BioProject, PRJNA828354 |
| Henault M, Marsit S, Charron G, Landry CR | 2019 | Ploidy instability in experimental hybrid populations | https://www.ncbi.nlm.nih.gov/bioproject/?term=PRJNA515073 | NCBI BioProject, PRJNA515073 |

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
