## [Editor Report · eLife assessment]

This **valuable** study advances our understanding of the forces that shape the genomic landscape of transposable elements. By exploiting both long-read sequencing of mutation accumulation lines and in vivo transposition assays, the authors offer **compelling** evidence that structural variation rather than transposition largely shapes transposable element copy number evolution in budding yeast. The work will be of interest to the transposable element and genome evolution communities.

---

## [Referee Report · Reviewer #1 (Public Review)]

Henault et al build on their own previous work investigating the longstanding hypothesis that hybridization between divergent populations can activate transposable element mobilization (transposition). Previously they created crosses of increasing sequence divergence, using both intra- and inter-species hybrids and passaged them neutrally for hundreds of generations. Their previous work showed that neither hybrids isolated from natural environments nor hybrids from their mutation accumulation lines showed consistent evidence of increased transposable element content. Here, they sequence and assemble long read genomes of 127 of their mutation-accumulation lines and annotate all existing and de novo transposable elements. They find only a handful of de novo transposition events, and instead demonstrate that structural variation (ploidy, aneuploidy, loss of heterozygosity) plays a much larger role in the transposable element load in a given strain. They then created transposable element reporter constructs using two different Ty1 elements from S. paradoxus lineages and measured transposition rate in a number of intraspecific crosses. They demonstrate that transposition rate is dependent on both the Ty1 sequence and the copy number of genomic transposable elements, the latter of which is consistent with what has been observed in the literature of transposable element copy number control in Saccharomyces. To my knowledge, others have not directly tested the effect of Ty1 sequence itself (have not created diverse Ty1 reporter constructs), and so this is an interesting advance. Finally, the authors show that mitotype has a moderate effect on transposition rate, which is an intriguing finding that will be interesting to explore in future work.

The authors state that their results from their current work support results taken from their previous study using short read sequencing data of the same lines. The argument that follows is whether the authors gained anything novel from long read sequencing. While major results did not change from their previous work, the addition of long read sequencing did provide novel insight into the comparison of de novo transposition and structural variation that was not possible with short read sequencing. Additionally, this allowed the authors to compare estimates of transposition from two methods (inferred from mutation accumulation lines and from reporter assays).

Overall, this study represents a large effort to investigate how genetic background can influence transposable element load and transposition rate. The long read sequencing, assembly, and annotation, and the creation of these reporter constructs is non-trivial. Their results are straightforward, well supported, and are a nice addition to the literature.

---

## [Referee Report · Reviewer #2 (Public Review)]

This is an interesting followup study that uses long read sequencing to examine previously constructed mutation accumulation lines between wild populations of *S. cerevisiae* and *S. paradoxus*. They also complement this work with reporter assays in hybrid backgrounds. The authors are attempting to test the hypothesis that hybridization leads to genome shock and unrestrained transposition. The paper largely confirms previous results (suggesting hybridization does not increase transposition) that are well cited and discussed in the paper, both from this group and from the Smukowski Heil/Dunham group but extends them to a new set of species/hybrids and with some additional resolution via the long read sequencing. The paper is well written and clear and I have no serious complaints.

In the abstract, the authors make three primary claims:

Structural variation plays a strong role in TE load.

Transposition plays only a minor role in shaping the TE landscape in MA lines.

Transposition rates are not increased by hybridization but are affected by genotype specific factors.

Comments on revised submission:

I found all three claims supported, albeit with some minor questions. Those questions were answered by the authors in revision. I appreciate the authors revisions and feel the paper is now in better shape than upon the original submission.

---

## [Author Response]

The following is the authors’ response to the original reviews.

**Public Reviews:**

**Reviewer #1 (Public Review):**
Henault et al build on their own previous work investigating the longstanding hypothesis that hybridization between divergent populations can activate transposable element mobilization (transposition). Previously they created crosses of increasing sequence divergence, using both intra- and inter-species hybrids, and passaged them neutrally for hundreds of generations. Their previous work showed that neither hybrids isolated from natural environments nor hybrids from their mutation accumulation lines showed consistent evidence of increased transposable element content. Here, they sequence and assemble long-read genomes of 127 of their mutation-accumulation lines and annotate all existing and de novo transposable elements. They find only a handful of de novo transposition events, and instead demonstrate that structural variation (ploidy, aneuploidy, loss of heterozygosity) plays a much larger role in the transposable element load in a given strain. They then created transposable element reporter constructs using two different Ty1 elements from S. paradoxus lineages and measured the transposition rate in a number of intraspecific crosses. They demonstrate that the transposition rate is dependent on both the Ty1 sequence and the copy number of genomic transposable elements, the latter of which is consistent with what has been observed in the literature on transposable element copy number control in Saccharomyces. To my knowledge, others have not directly tested the effect of Ty1 sequence itself (have not created diverse Ty1 reporter constructs), and so this is an interesting advance. Finally, the authors show that mitotype has a moderate effect on transposition rate, which is an intriguing finding that will be interesting to explore in future work.This study represents a large effort to investigate how genetic background can influence transposable element load and transposition rate. The long read sequencing, assembly, and annotation, and the creation of these reporter constructs are non-trivial. Their results are straightforward, well supported, and a nice addition to the literature.The authors state that the results from their current work support results taken from their previous study using short-read sequencing data of the same lines. The argument that follows is whether the authors gained anything novel from long-read sequencing. I would like to see the authors make a stronger argument for why this new work was necessary, and a more detailed view of similarities or differences from their previous study (when should others choose to do long read vs. short read of evolved lines?).

We thank the reviewer for the suggestion. While we initially aimed to justify the relevance and novelty of the current in relation to our previous study, we understand that this justification may not have been strong enough.

In the second paragraph of the introduction, we explain how the multidimensional nature of TE load makes it more complex to characterize that simply reporting the abundance of a given TE family in a given genome. We added the following concluding sentence to further emphasize the importance of long reads in TE-focused genome inference:

“As such, ongoing technological and computational advances in genome inference, including long-read sequencing, will certainly be key to getting a detailed understanding of the dynamics of TEs and the underpinning evolutionary forces.”

In the penultimate introductory paragraph, we summarize our previous work from 2020 and highlight that the evolution of Ty contents in MA lines was inferred from aggregate measures of genomic abundance of TE families using short reads. We then make the point that combinations of multiple SVs could affect the landscape of TEs in ways that are not reflected by crude short-read measures. We added the following sentence to further emphasize this point and contrast it with the necessity of using more powerful methodologies for genome resolution:

“Under this scenario, measuring Ty family abundance would yield no significant net change, and the dissection of the underlying SVs using short reads could often be challenging.”

Relatedly, the authors should report the rates of structural variants that they observe. How are these results similar/different from other mutation-accumulation work in *S. cerevisiae*?

Since this work does not attempt to provide an exhaustive report of all the SVs in the MA lines, but rather focus on attributing an SV type to individual loci occupied by TEs, we cannot include these estimates, excepted for de novo transposition itself (see below). We added the following sentence to the Results section on the classification of Ty loci by SV types:

“We note that the current methodology does not aim at providing an exhaustive quantification of all SVs in the MA lines, as previously done for some SV types (Marsit et al., 2021), but focuses solely on loci containing Ty elements.”

We added estimates of the average retrotransposition rate in the MA experiment based on the number of de novo insertions detected in the MA lines genomes.

Figure 4:

“The average retrotransposition rates estimated from the counts of de novo insertions (per line per generation per element) are the following: CC1, 1.0✕10-5; CC2, 4.9✕10-6; CC3, 7.6✕10-6; BB1, 1.5✕10-5; BC2, 1.7✕10-5; BA1, 6.5✕10-6; BA2, 2.2✕10-5; BSc1, 3.6✕10-5.”

We added the following paragraph in the Discussion section to specifically discuss these estimates in relation to the in vivo measurements.

“We note that while the CC crosses tend to have the lowest retrotransposition rates as estimated from the de novo insertions (~1✕10-5 per line per generation per element; Figure 4), these values are several orders of magnitude higher than the in vivo measures in SpC backgrounds. The discrepancy between these estimates could be due to uncharacterized biases inherent to each method. They could also be linked to differences between the parental genotypes used to generate the MA crosses and the fluctuation assays. One major difference is the use of ade2 genotypes in the MA parents, a strategy that was initially adopted to provide a marker for the loss of mitochondrial respiration (Joseph and Hall, 2004; Lynch et al., 2008). It has been shown that the induction of adenine starvation through minimal adenine concentration in the medium and deletion of ADE2, which inactivates the adenine de novo biosynthesis pathway, increases Ty1 transcript levels (Todeschini et al., 2005), resulting in higher transposition rates. Rich complex medium like the one that was used for the MA experiment (YPD) can exhibit substantial variation in adenine concentration (VanDusen et al., 1997), and adenine can quickly become the limiting nutrient for ade2 strains (Kokina et al., 2014). Thus, we cannot exclude that the choice of initial ade2 genotypes could have inflated the transposition rates in the MA experiment.”

Since the authors show a small, but consistent influence of mitotype on transposition rates, adding further evidence for the role of mtDNA in regulating transposition, I'm curious what the transposition rate of a p0 strain is. I think including these results could make this observation more compelling.

We agree that measuring in vivo transposition rates in ρ0 backgrounds would be an interesting avenue. However, there is a large distinction between having non-functional mitochondrial respiration in ρ0 strains and inheriting diverse functional mtDNA haplotypes. The effects we show are all linked to the reciprocal inheritance of intact mtDNAs, producing ρ+ strains that are all respiration-competent, as shown by our growth confirmations on non-fermentable carbon sources for all the diploid backgrounds generated. While potentially interesting, adding transposition rates measures for the ρ0 backgrounds seems hard to justify in the context of our results.

**Reviewer #2 (Public Review):**
This is an interesting follow-up study that uses long-read sequencing to examine previously constructed mutation accumulation lines between wild populations of *S. cerevisiae* and *S. paradoxus*. They also complement this work with reporter assays in hybrid backgrounds. The authors are attempting to test the hypothesis that hybridization leads to genome shock and unrestrained transposition. The paper largely confirms previous results (suggesting hybridization does not increase transposition) that are well cited and discussed in the paper, both from this group and from the Smukowski Heil/Dunham group but extends them to a new set of species/hybrids and with some additional resolution via the long read sequencing. The paper is well written and clear and I have no serious complaints.In the abstract, the authors make three primary claims:Structural variation plays a strong role in TE load.Transposition plays only a minor role in shaping the TE landscape in MA lines.Transposition rates are not increased by hybridization but are affected by genotype-specific factors.I found all three claims supported, albeit with some minor questions below:Structural variation plays a strong role in TE load.Convinced of this result. However:Line 185-187/Figure 3C: I'm curious given that the changes in Ty count are so often linked to changes in gross DNA sequence whether the count per total DNA sequence is actually changing on average in these genomes. Ie., does hybridization tend to increase TE count via CNV or does hybridization tend to increase DNA content in the MA lines and TEs come along for the ride?

The Ty content definitely “rides along” with the rest of the genome that is affected by retrotransposition-unrelated SVs. To further highlight this point, we added a panel (E) to Figure 3 in which we correlate the net Ty copy number change (same as panel D, formerly C) to the corresponding genome size, which reflects the amount of DNA lost/gained by all SV types. We added the following to the results section:

“The distributions of net Ty CN change per MA line showed that most crosses had significant gains (Figure 3D), suggesting that Ty load can often increase as a result of random genetic drift. Some (but not all) of these crosses also exhibited significant increases in genome size after evolution (Supplemental Figure S7A). The net Ty CN changes per MA line subgenome were globally correlated to the corresponding changes in subgenome size (Figure 3E). Even after excluding polyploid lines (which have the largest changes in both Ty CN and genome size), we found a significant relationship between the two variables (mixed linear model with random intercepts and slopes for MA crosses, P-value=3.71✕10-9; Supplemental Figure S7B), indicating that SVs affecting large portions of the genome have a substantial impact on the Ty landscape.”

One question about ploidy (lines 175-177):Both aneuploidy and triploidy seem easy to call from this data. A 3:1 tetraploidy as well. However, in Figure 2B there are tetraploids that are around the 1:1 line. How are the authors calling ploidy for these strains? This was not clear to me from the text.

This detail was indeed missing from the manuscript. The ploidy level of all MA lines was previously measured by DNA staining and flow cytometry, and the ploidy level of the subgenomes of each polyploid MA line was previously inferred from short-read sequencing. We modified the figure captions and the main text to include this along with the corresponding references:

Figure 2:

“The ploidy level of each line was previously determined by DNA staining and flow cytometry (Charron et al., 2019; Marsit et al., 2021).”

Main text:

“The ratio of classified bases per subgenome was consistent with the corresponding ploidy levels: triploid BC lines had two copies of the SpC subgenome, while tetraploid lines had both SpC subgenomes duplicated (Charron et al., 2019; Marsit et al., 2021) (Figure 2B).”

“Finally, we used the ploidy level of each MA line subgenome as previously measured by flow cytometry and short-read sequencing (Charron et al., 2019; Marsit et al., 2021).”

**Reviewer #3 (Public Review):**
Henault et al. address the important open question of whether hybridization could trigger TE mobilization. To do this they analysed MA lines derived from crosses of *Saccharomyces paradoxus* and *Saccharomyces cerevisiae* using long-read sequencing. These MA lines were already analysed in a previous publication using Illumina short-read data but the novelty of this work is the long-read sequencing data, which may reveal previously missed information. It is an interesting message of this study that hybridization between the two species did not lead to much TE activity. Due to this low activity, the authors performed an additional TE activity assay in vivo to measure transposition rates in hybrid backgrounds. The study is well written and I cannot spot any major problems. The study provides some important messages (like the influence of the genotype and mitochondrial DNA on transposition rates).Major commentsWhat I miss the most in this work is the perspective of the host defence against TEs in *Saccharmoces*. Based on such a mechanistic perspective, why do the authors think that hybridization could lead to a TE reactivation? For example, in *Drosophila* small RNAs important for the defence against a TE, are solely maternally transmitted. Hybrid offspring will thus solely have small-RNAs complementary to the TEs of the mother but not to the TEs of the father, therefore a reactivation of the paternal TEs may be expected. I was thus wondering, what is the situation in yeast. Why would we expect an upregulation of TEs? Without such a mechanistic explanation the hypothesis that TEs should be upregulated in hybrids is a bit vague, based on a hunch.

We agree with the reviewer that in the first version of the manuscript, the justification for the investigation of the reactivation hypothesis in the first place was not self-sufficient and relied too much on our previous work, upon which this article builds. We extensively remodeled the introduction to better justify the investigation of this hypothesis in the context of the current knowledge on the regulation of Ty elements in Saccharomyces.

**Reviewer #1 (Recommendations For The Authors):**
It's interesting that the net change in transposable element copy number in mutation accumulation lines is either insignificant or gain, and never a significant loss. I think this could make a nice discussion point regarding the roles of drift and selection on TE load.

We thank the reviewer for the suggestion and agree that this is an interesting perspective that we did not explore in the first version of the manuscript. We thus included a short discussion point in theResults:

“The distributions of net Ty CN change per MA line showed that most crosses had significant gains (Figure 3D), suggesting that Ty load can often increase as a result of random genetic drift.”

We also added the following paragraph to the discussion section:

“Our experiments illustrate how under weakened natural selection efficiency, TE load can increase in hybrid genomes by the action of transposition-unrelated SVs. This offers a nuanced perspective on the classical interpretation of the transposition-selection balance model (Charlesworth et al., 1994; Charlesworth and Langley, 1989), in which increased TE load would be predominantly driven by the relaxation of purifying selection against TE insertions generated by de novo transposition. Our results suggest that SVs arising in the context of hybridization can act as a significant source of TE insertion polymorphisms which natural selection can purge more or less efficiently, depending on the population genetic context. This is closely related to the idea that sexual reproduction could favor the spread of TE families, contributing to their evolutionary success (Hickey, 1982; Zeyl et al., 1996). Since the insertion polymorphisms that contribute to increase TE load mostly originate from standing genetic variation, they could be less deleterious and thus harder for natural selection to purge efficiently.”

The point about the role of LOH in TE load is cool!

We thank the reviewer for their enthusiasm, it is one of our favorite results as well.

Figure 1: Add a figure component of the green box and label it Ty1 or TE.

We modified Figure 1 accordingly.

Figure 2C: what is the assembly size ratio?

We added the following sentence to the figure caption to clarify what we define as assembly size ratio:

“Assembly size ratio refers to the ratio of subgenome assembly size to the corresponding parental assembly size.”

Something cut off in the N50 plot axis

Unfortunately, we can’t seem to understand what the reviewer meant with this comment, nothing seems cut out of the figure panel 2C in any of our versions of the manuscript.

**Reviewer #2 (Recommendations For The Authors):**
These are all minor comments/suggestions that the authors can take or leave.Line 42: "fuels" should be "fuel".

Since the verb refers to “source” and not “variants”, we believe it should be at the third person singular.

Line 43: unclear what the authors mean by "regroup".

We understand how this phrasing may sound strange. We modified the sentence accordingly:

“Structural variation is a term that encompasses a broad variety of large-scale sequence alterations”

Line 51-52: There are a couple of really nice papers that could be cited here from Anna Selmecki's group (Todd et al. 2020, Todd and Selmecki 2019, both in eLife).

We thank the reviewer for the suggestions, we included some of these references in the manuscript.

Figure 1: This is a nice cartoon! I'd suggest spelling out LOH here for a truly naive reader.

We modified the Figure 1 accordingly.

Figure 3A: One thing that is slightly lost here in the presentation is the relative frequency of the different events because of the changing scales across 3A. I can see why you want to do it this way, but would consider whether there may be a way to present this that makes it more obvious how much more frequent polyploidy is than excision for example.

We agree with the reviewer that the focus of this visualization is to compare crosses and individual MA lines within SV types, and fails to display the relative importance of each SV type. We solved this by including an additional panel (new 3A) that shows how the number of Ty loci affected by each SV type scales in comparison to others.

Figure 5: I'm not a fan of the gray bars highlighting the individual strains. This made the graph less intuitively readable for me.

We tend to agree with the reviewer and rolled back to a previous version of Figure 5 that was lighter on annotations.

One thing I would like to see in the future from this data (definitely not in this paper) is genome rearrangements within these hybrid MA lines. How often are there structural changes and how often are those changes mediated by repeats including TEs?

We completely agree with the reviewer that this would be a very interesting avenue, with a distinct (and likely higher) set of challenges at the analysis level compared to simply focusing on TE sequences like we did here. We hope to be able to tackle this goal in the future of this project.

**Reviewer #3 (Recommendations For The Authors):**
I'm not from the yeast field. But why this focus on the Ty-load? Are Ty's the only active TEs in yeast? Provide some background on the TE landscape in yeast and a justification for focusing on Ty's.

We agree with the reviewer that this point was only implicit in the introduction. We modified the introductory segment on Saccharomyces yeasts to mention that Ty retrotransposons are the only TEs found in these genomes, thus explaining the exclusive focus on them. It now reads as follows:

“In the case of *Saccharomyces cerevisiae*, the only TEs found are five families of long terminal repeat (LTR) retrotransposons families named Ty1-Ty5 (Kim et al., 1998).”

56 I would argue that Petrov et al 2003 is not the best citation for arguing that TEs can lead to genomic rearrangement through ectopic recombination. Petrov solely showed that some long TE families are at lower population frequency than short TE families ones. This could be due to many reasons (e.g. recent activity of long TEs - mostly LTRs) but Petrov interpreted the data as being due to ectopic recombination. Petrov, therefore, did not demonstrate any direct evidence for the involvement of ectopic recombination.

We agree with the reviewer that this reference is not the best choice to simply support the role of TEs in generating ectopic recombination events and modified the references accordingly.

For the assembly the authors used two steps (1) separate the reads based on similarity to a subgenome (2) and assembly the reads from the resulting two sets separately. This is probably the only viable approach, but I'm wondering if this step can lead to some biases (many reads may not be assigned to one sub-genome or assigned to the wrong sub-genome). An alternative, possibly less biased approach, would be to use one of the emerging assemblers that promise to assemble sub-genomes. Maybe discuss why this approach was not pursued.

We completely agree that our method has some level of bias. We adopted it because it seemed the most appropriate to answer our question, which required to resolve individual TE insertions at the level of single haplotype sequences. One specific challenge of this dataset is that we have a relatively wide range of nucleotide divergence between parental subgenomes in the different MA crosses, from <1% to ~15%. The efficiency of haplotype separation from tools that are not necessarily designed to be tunable with respect to the level of nucleotide divergence seemed uncertain, which is why we opted for a custom methodology. Although read non-classification remains a problem that is hard to solve (and would remain so using orthogonal strategies), we believe that read misclassification is minimized by our stringent criteria for read classification. The goal of this study was not to develop a tool nor to benchmark our approach against existing diploid assembly tools. It yielded phased genome representations that were of sufficient completeness and contiguity to confidently answer our questions, and we believe that pushing the discussion towards technical considerations would fall outside of our main objective.

The authors used a decision tree to classify Ty loci. What were the training data? How were the trees validated? Decision tree is a technical term for a classifier in machine learning. I do not think the authors used machine learning in this work, but rather an "an ad-hoc set of rules". The term decision tree in this study is misleading.

We believe that the term “decision tree” can simply refer to a hierarchy of conditional rules implemented as a classification algorithm. As the reviewer pointed, it is clear from the manuscript that none of the analyses performed include any form of training or fitting of a machine learning classifier. However, we agree that its specific reference to the machine learning classifier can create unnecessary confusion. We thus agree to remove this term from the manuscript and replaced all its instances by “a hierarchy of binary rules”.

272: as it is the CNC explanation does not make a lot of sense to me; some information is missing, is p22 expression increasing with copy numbers?

Yes, p22 expression correlates positively with the CN of p22-expressing Ty1 elements.

Why are the two alternative downstream codons important?

We thought it would be useful to mention the two start codons at this point because later in the discussion, we bring the conservation of the first start codon as an observation consistent with the putative expression of p22 in S. paradoxus. We also thought that it helped clarify the mechanism by which the N-truncated version of the protein is expressed.

p22 interferes with assembly viral particles when in high copy numbers, but what happens when at low copy numbers, is it essential for retroviral activity? Is it even necessary for the virus or just some garbage product (they mention N-truncated).

To our knowledge, these questions regarding the potential molecular functions of p22 outside of a retrotransposition restriction factor are still open. We added details to the background on CNC in the Introduction and Results section to help clarify some the points raised:

Introduction:

“The best known regulation mechanism in yeast is termed copy number control (CNC) and was characterized in the Ty1 family of *S. cerevisiae*. This mechanism is a potent copy-number dependent negative feedback loop by which increasing the CN of Ty1 elements strengthens their repression (Czaja et al., 2020; Garfinkel et al., 2003; Saha et al., 2015).”

Results:

“The mechanism of negative copy-number dependent self-regulation of retrotransposition (CNC) was characterized in the Ty1 family of *S. cerevisiae* (Garfinkel et al., 2016). This mechanism relies on the expression of an N-truncated variant of the Ty1 capsid/nucleocapsid Gag protein (p22) from two downstream alternative start codons (Nishida et al., 2015; Saha et al., 2015). p22 expression scales up with the CN of Ty1 elements that encode it (Tucker et al., 2015), which gradually interferes with the assembly of the viral-like particles essential for Ty1 replication (Cottee et al., 2021; Saha et al., 2015). Thus, CNC yields a steep negative relationship between the retrotransposition rate measured with a tester element and the number of Ty1 copies in the genome (Garfinkel et al., 2003; Tucker et al., 2015).”

mtDNA influences transposition, is anything known about the mechanism?

When presenting this result, we make it clear that this finding is not new and was previously observed in *S. cerevisiae* x *S. uvarum* hybrids by Smukowski-Heil et al. (2021). In this reference, the authors discuss multiple mechanisms by which mitochondrial biology and mito-nuclear interplay may affect transposition rate, although their data cannot support one specific hypothesis. Our data does not to allow to further dissect the mechanistic basis of the mtDNA effect, not more than the effect of distinct Ty1 natural variants. Since we simply provide new independent evidence for the mtDNA effect, it seems to us that repeating the discussion on putative mechanisms while bringing no support to any given hypothesis would be of limited relevance.

During the first reading, I got quite confused about what CN means (copy number as it turned out). I suggest using abbreviations only if absolutely necessary, and I'm not entirely convinced it is necessary here. But I leave this to the discretion of the authors.

We agree that the excessive use of abbreviations in manuscripts is annoying. However, in this case, “copy number” is used so extensively that its abbreviation seemed to improve the reading experience. Thus, we would prefer to keep it unchanged.

Fig 3D: Wilcoxon Rank sum test. It is not clear to me what was tested here? Which data were used?

We confirm that the statistical test employed is the Wilcoxon signed-rank test, and not the Wilcoxon rank-sum test (also known as Mann-Whitney U-test). The Wilcoxon signed-rank test is used here as a non-parametric one-sample test against the null hypothesis that the distribution is centered around zero.

de novo -> italics

We choose to follow the recommendation of the general style conventions of the ACS guide for scholarly communications not to italicize common Latin terms like “de novo”, “e.g.” and “i.e.”.